# A co-crystal berberine-ibuprofen improves obesity by inhibiting the protein kinases TBK1 and IKKε

Man Wang[1], Rong Xu[1], Xiaoli Liu[1], Ling Zhang[1], Siyan Qiu[1], Yuting Lu[1], Peng Zhang[1], Ming Yan[1] & Jing Zhu [1,2]✉

Berberine (BBR) exerts specific therapeutic effects on various diseases such as diabetes, obesity, and other inflammation-associated diseases. However, the low oral bioavailability (below 1%) of berberine due to its poor solubility and membrane permeability limits its clinical use. In this paper, we have prepared a 1:1 co-crystal berberine-ibuprofen (BJ) using drug salt metathesis and co-crystal technology. Pharmacokinetic studies demonstrate a 3-fold increase in vivo bioavailability of BJ compared to that of BBR, and BJ is more effective in treating obesity and its related metabolism in vitro and in vivo. We also find that BJ promotes mitochondrial biogenesis by inhibiting TBK1 and inducing AMP-activated protein kinase (AMPK) phosphorylation, and BJ increases adipocyte sensitivity to catecholamine by inhibiting IKKε. Together, our findings support that co-crystal BJ is likely to be an effective agent for treating obesity and its related metabolic diseases targeting TBK1 and IKKε.

[1] Jiangsu Key Laboratory for Pharmacology and Safety Evaluation of Chinese Materia Medica, School of Pharmacy, Nanjing University of Chinese Medicine, Nanjing, China. [2] Department of Neurology and Neuroscience, Johns Hopkins School of Medicine, Baltimore, MD, USA. ✉email: 830640@njucm.edu.cn

Obesity has become a global public health problem and can cause complications and even death[1–6]. Obesity develops from a chronic imbalance between energy intake and energy expenditure[7,8]. In addition, growing evidence indicates that obesity is closely associated with low-grade chronic inflammation, which induces insulin resistance, the central pathological mechanism underlying type 2 diabetes (T2D)[9–12]. Therefore, changing metabolic efficiency, increasing energy consumption of vital metabolic organs (such as adipose tissue), and knocking out or suppressing inflammatory pathways with drugs to disrupt the link between genes or diet-induced obesity and insulin resistance are critical alternative treatment strategies.

Obesity-induced chronic inflammation increases the production of proinflammatory cytokines, such as TNFα, in adipose tissue, and these cytokines stimulate NF-κB activity in adipocytes and induce TBK1 and IKKε expression[13–16]. Recent clinical studies have also shown that IKKε/TBK1 inhibitor can improve hepatic steatosis and insulin sensitivity in obese patients with non-alcoholic fatty liver disease (NAFLD) and T2D[17]. Activated TBK1 can phosphorylate AMPKα directly[15]. AMPK is a crucial regulator of the mechanisms that control energy expenditure. AMPK can sense the energy state and respond by increasing lipid oxidation and mitochondrial biogenesis while reducing fat and glycogen production. TBK1 reduces lipid oxidation and mitochondrial biogenesis by directly inhibiting AMPK activity in adipocytes. Adipocyte-specific TBK1 deficiency increases energy expenditure and reduces HFD-induced obesity[15]. However, IKKε does not affect AMPK phosphorylation[15]. IKKε reduces the level of cAMP in adipocytes and inhibits camp-mediated β-adrenergic signal transduction, which induce catecholamine resistance, thus reducing lipolysis and thermogenesis[14]. IKK-epsilon (Ikbke) gene knockdown restores catecholamine sensitivity, with subsequent increases in uncoupling protein 1(UCP1) expression and thermogenesis[14]. In addition, mice lacking IKKε also display improvements in insulin sensitivity and glucose and lipid homeostasis, as well as reductions in the activation of chronic inflammation pathways[18]. Using dual-specificity inhibitors of IKKε and TBK1 can increase energy consumption and insulin sensitivity by inducing fat cell browning and enhancing catecholamine sensitivity[13,17]. The activation of IKKε and TBK1 can reduce energy consumption and increase energy storage and insulin resistance. Therefore, inhibiting IKKε and TBK1 can ameliorate obesity, and its inflammation, and alleviate obesity-related metabolic disorders. Currently, traditional medicines are often used to treat obesity. However, they frequently have unacceptable side effects that limit the application of these medicines. There is an urgent need for safe, effective, and readily available drugs to treat obesity.

Berberine (BBR), a natural isoquinoline alkaloid compound, is commonly used as an over-the-counter (OTC) drug to treat gastrointestinal tract infections. Recently, numerous studies have shown that BBR exerts beneficial biological effects on obesity and insulin resistance[19–22]. Treatment of db/db mice and HFD fed Wistar rats with BBR reduces the fat mass and improves glucose homeostasis and insulin sensitivity[23]. Further research shows that BBR regulates white adipose tissue (WAT) browning by activating AMPK and inducing peroxisome proliferator-activated receptor γ coactivator-1α (PGC-1α) dependent mitochondrial biogenesis in db/db mice, but BBR has no effect on brown adipose tissue (BAT) thermogenesis in adipose tissue-specific AMPKα1 and AMPKα2 knockout mice[19,22]. BBR can also inhibit lipid accumulation via the phosphorylation of Thr172 in the α-subunit of AMPK and acetyl-CoA carboxylase (ACC) in 3T3-L1 adipocytes[23]. Moreover, studies of individuals with obesity and T2D reported that oral BBR (1000 mg/d or 1500 mg/d) administration could reduce body weight, inflammation, fasting plasma glucose, and improve insulin sensitivity[22,24,25]. These findings suggest that BBR is a promising drug (active ingredient) for treating obesity.

However, pharmacokinetic (PK) modeling studies showed that low plasma concentrations of BBR are observed after oral administration due to its extensive metabolism in the body. Only less than 5% of the oral dose of BBR enters the systemic circulation in human subjects[26]. After oral administration of BBR (300 mg/d) for seven days, the steady-state concentration in human plasma is approximately 0.3 ng/ml, and the maximum plasma concentration ($C_{max}$) is approximately 0.4 ng/ml in acute oral BBR (400 mg) provocation tests[26]. The absolute bioavailability of BBR in rats is low, only 0.68%[27]. The low oral bioavailability has greatly limited its more comprehensive application. BBR chloride dihydrate (BCl·2H$_2$O, hereafter referred to as BBR for simplicity) is the most common salt form of BBR, but the bioavailability of this form is still limited. This study used drug salt metathesis and co-crystal technology to overcome the low-fat solubility and bioavailability of existing forms of BBR. The drug co-crystal refers to the crystal of the active pharmaceutical ingredient (API) and co-crystal former (CCF) bonded by a non-covalent bond[28–30]. Drug co-crystal is a good choice to make for the deficiency of drug salt, which improve API's physicochemical and biopharmaceutical properties[31–37]. On the one hand, the original chemical structure of the drug is preserved, and on the other hand, the drug's stability, solubility, and bioavailability are improved. When other APIs replace CCF, it can increase the pharmacological activity of the original drug[38–40]. Our laboratory's previous BBR salt and crystal screening experiments showed that BBR can form co-crystal with ibuprofen (Ibu). BBR does not contain abundant hydrogen bond donors and receptors, so creating hydrogen bonds with other compounds is difficult. However, during the characterization of the new co-crystal of BBR synthesized in this study, we found that this co-crystal structure is formed partly due to the presence of crystalline water. The observed phenomenon of water molecules participating in the formation of co-crystal also has also been noted in previous studies, and this may be explained by the relationship between deliquescence and co-crystal[41,42]. In addition, the crystal form, fat solubility, and bioavailability of the prepared berberine-ibuprofen drug co-crystal (BJ) were compared with those of BBR. The oral bioavailability of BJ increased by threefold compared with that of BBR. We also evaluated the inhibitory effect of the crystalline compound BJ on obesity and explored the related mechanisms.

## Results

**BJ increases the bioavailability and dynamic solubility compared to BBR.** After a series of solvent synthesis methods were screened, BJ crystals were successfully generated. The BJ had a uniform long rod-like morphology according to polarized light microscope (PLM) observation (Fig. 1a). First, we performed Nuclear Magnetic Resonance (NMR) experiments to characterize the composition of BJ. The pattern shows that the BJ structure contained both 1:1 BBR and Ibu (Fig. 1b, Supplementary Fig. 1a–c).[1]H NMR data of BJ:[1]H NMR (500 MHz, MeOD) δ 9.79 (s, 1H), 8.72 (s, 1H), 8.14 (d, J = 9.1 Hz, 1H), 8.02 (d, J = 9.1 Hz, 1H), 7.68 (s, 1H), 7.27 (d, J = 8.0 Hz, 2H), 7.10–6.90 (m, 3H), 6.13 (s, 2H), 4.18 (d, J = 48.5 Hz, 6H), 3.55 (q, J = 7.1 Hz, 1H), 3.33 (dt, J = 3.3, 1.6 Hz, 4H), 3.30–3.19 (m, 2H), 2.41 (d, J = 7.2 Hz, 2H), 1.87–1.76 (m, 1H), 1.40 (d, J = 7.1 Hz, 3H), 0.89 (d, J = 6.6 Hz, 6H).

Then the results of differential scanning calorimetry (DSC) and thermogravimetric analysis (TGA) experiments showed that the melting point of BJ was approximately 152 °C (Fig. 1c), which was different from the respective melting points of BBR and Ibu

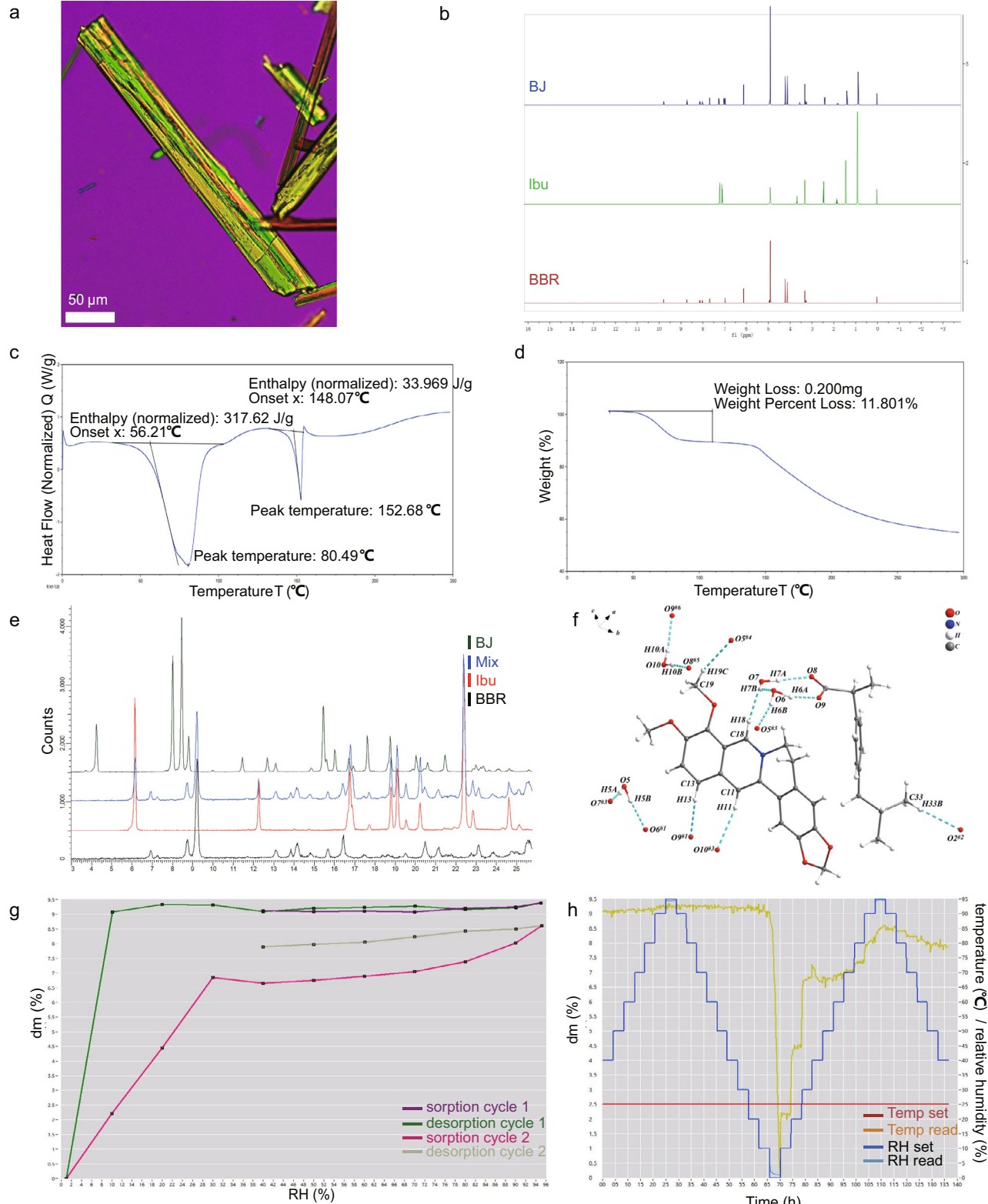

**Fig. 1 BJ is characterized as a co-crystal tetrahydrate of BBR and Ibu. a** The images observed under the polarizing microscope of BJ. **b** Comparative [1]H NMR hydrogen spectra of BBR, Ibu and BJ. **c**, **d** DSC (**c**) and TGA (**d**) curves of BJ. **e** The PXRD patterns of BBR, Ibu, Mix (BBR + Ibu) and BJ. **f** Molecular structure of BJ, showing the atom-labeling scheme (H atoms are shown as small white spheres; O atoms are shown as small red spheres; hydrogen bonds are represented by light blue dotted line). **g**, **h** DVS curves of BJ.

**Table 1 Crystal data and structure refinement for BJ.**

| | |
|---|---|
| Empirical formula | $C_{20}H_{18}NO_4 \cdot C_{13}H_{17}O_2 \cdot 4(H_2O)$ |
| Formula weight | 613.68 |
| Temperature (K) | 120(2) |
| Crystal system | Orthorhombic |
| Space group | $Pna2_1$ |
| $a$ (Å), $b$ (Å), $c$ (Å) | 11.4018(4), 41.2164(16), 6.6951(3) |
| $\alpha$ (°), $\beta$ (°), $\gamma$ (°) | 90, 90, 90 |
| $V$ (Å$^3$) | 3146.3(2) |
| $Z$ | 4 |
| $D_c$ (g·cm$^{-3}$) | 1.296 |
| $\mu$ (mm$^{-1}$) | 0.789 |
| $F$ (000) | 1312 |
| Crystal size (mm$^3$) | 0.11 × 0.04 × 0.02 |
| $\theta$ Range | 4.023–74.491 |
| Reflections collected | 30297 |
| Independent reflections | 6150 [$R_{int} = 0.0912$] |
| Reflections observed [$I > 2\sigma(I)$] | 4817 |
| Data/restraints/parameters | 6150/2/430 |
| Goodness-of-fit on $F^2$ | 1.048 |
| $R/wR$ [$I > 2\sigma(I)$] | 0.0634/0.1659 |
| $R/wR$ (all data) | 0.0812/0.1845 |
| Max., Min. $\Delta\rho$ (e·Å$^{-3}$) | 0.886, −0.316 |

(Supplementary Fig. 2a, b). The dehydrated phase appeared in the plot, and the 11.8% weight loss suggested that BJ might contain four molecules of crystalline water (Fig. 1d). Additionally, the results of powder X-ray diffraction (PXRD) experiments showed that the diffraction peaks of BJ different from those of the raw materials (Fig. 1e). These characterizations indicate the creation of new crystalline forms.

Next, we immediately determined the molecular conformation of BJ by single crystal X-ray diffraction (SCXRD) experiments. The experimentally simulated PXRD maps were of the same crystal form as the one used in the previous PXRD experiments (Supplementary Fig. 3). The SCXRD experimental results showed that BJ is a co-crystal formed by BBR and Ibu, which are connected by hydrogen bonds (Fig. 1f, Supplementary Figs. 4 and 5). The crystallographic data and refinement details of BJ are summarized in Table 1. SCXRD analysis shows that the BJ crystallizes in Pna21 space group, with one molecule of BBR, one molecule of Ibu and four molecules of water in an asymmetric unit. The maintenance of this structure relied on a variety of hydrogen bonds (Supplementary Table 1). Water molecules formed a bridge between the hydrogen on the quinoline ring of BBR and the oxygen on the carboxyl group of Ibu. Simultaneously, the electrostatic attraction between the anion, after the proton loss of Ibu, and the quaternary ammonium salt cation of BBR may also contribute to stabilizing the co-crystal structure.

Finally, we performed a dynamic vapor adsorption (DVS) experiment to evaluate the relationship between BJ stability and humidity. The experimental results showed that when humidity increased from 40% to 95% and subsequently decreased to 10%, the weight of BJ did not change significantly. When the humidity was below 10%, BJ appeared partially dehydrated. However, when the humidity was increased above 30%, the dehydrated material reverted to the crystalline form of BJ again (Fig. 1g, h and Supplementary Fig. 6).

To initially assess whether BJ is better than BBR in drug delivery, we simulated a variety of biological fluids in vitro to measure their solubility and compare them (Fig. 2a, b), and the pH changes of the buffer before and after the test are listed in Supplementary Table 3. The experimental results showed that BBR of BJ showed higher solubility than BBR in the first 2 h of incubation in simulated gastric fluid (SGF), fasted state simulated

intestinal fluid -V1 (FaSSIF-V1) and fed state simulated intestinal fluid -V1 (FeSSIF-V1) (Fig. 2a). Additionally, in CMC-Na, BBR of BJ displayed greater solubility than BBR (Supplementary Fig. 7a, b). This phenomenon may occur due to the effect of the carboxyl group in Ibu. The protons on the carboxyl group of Ibu in the BJ structure are lost, resulting in an increase in the polarity of the entire structure, and thus an increase in solubility in some buffers. BJ exhibits better solubility in biological fluids in vitro, and this phenomenon also corresponds to the subsequent pharmacokinetic experiments showing better bioavailability than BBR.

The concentration-time profiles of BBR mother nucleus after administration of BBR and BJ suspensions by gavage are shown in Fig. 2c. The pharmacokinetic parameters were calculated by the statistical moment method of non-atrioventricular model. DAS 3.2.7 pharmacokinetic software was used to analyze the BBR and BJ concentration data. The main pharmacokinetic parameters obtained are shown in Supplementary Table 4.

Under the same dosage, the relative bioavailability of BJ was 308.5% according to the formula: $F = AUC_{(0 - \infty) BJ}/AUC_{(0 - \infty) BBR} * 100\%$. According to the data analysis described above, the peak time of BJ was prolonged to 3.8 h after the same dose of BBR was administration to rats, and the peak concentration was approximately 1.8 times of BBR. The relative bioavailability of BBR was 308.5%. The results showed that BJ promoted the oral absorption of BBR and significantly enhanced the bioavailability of BBR.

**BJ reduces the bodyweight of db/db mice and increases energy expenditure.** To investigate the effects of BJ on metabolism in vivo, we used db/db mice to model obesity and T2D. After 5 weeks, the db/db mice treated with BJ and BBR had lower body weight than the db/db mice treated with the vehicle. Notably, on day 35, the average body weight gain was 9.4 ± 2.7 g in the BJ (75 mg/kg)-treated group and 5.2 ± 0.9 g in the BJ (150 mg/kg)-treated group, while the BBR (150 mg/kg)-treated group had an average body weight gain of 7.6 ± 0.8 g. The effect of the BJ treatment on body weight was significantly enhanced compared with the same dose of BBR (Fig. 3a–c). However, food intake did not significantly differ among these groups (Fig. 3d). We also did not observe any significant difference in body weight in the Ibu (150 mg/kg)-treated mice (Supplementary Fig. 9a–c). As observed in the experiment, weight loss was not related to food intake in this study. We used the Comprehensive Lab Animal Monitoring System (CLAMS) to measure the energy expenditure of the db/db mice to investigate the causes of weight loss. CLAMS analysis showed that oxygen consumption (VO$_2$) and carbon dioxide elimination (VCO$_2$) levels in the BJ- and BBR -treated groups were significantly higher than those in the vehicle-treated group during the light-dark cycle (Fig. 3e–h). In contrast, the respiratory exchange ratio (RER) (Fig. 3i) in the BJ- and BBR -treated groups were decreased. Treatment with BJ (150 mg/kg) was more effective than treatment with BJ (75 mg/kg) or BBR (150 mg/kg). Notably, neither BJ nor BBR changed the spontaneous physical activity of the db/db mice under these conditions (Fig. 3j, k). Here, we noticed that treatment with BJ or BBR for 5 weeks reduced fasting serum ASAT, ALAT, BUN, and Cr levels in the db/db mice compared with the vehicle control (Supplementary Fig. 8a–d). These results suggest that both BJ and BBR treatment reduced weight and increased energy consumption of db/db mice, but treatment with 150 mg/kg BJ was more effective than BBR (150 mg/kg) and BJ (75 mg/kg). In addition, the weight loss of the db/db mice was not due to toxicity.

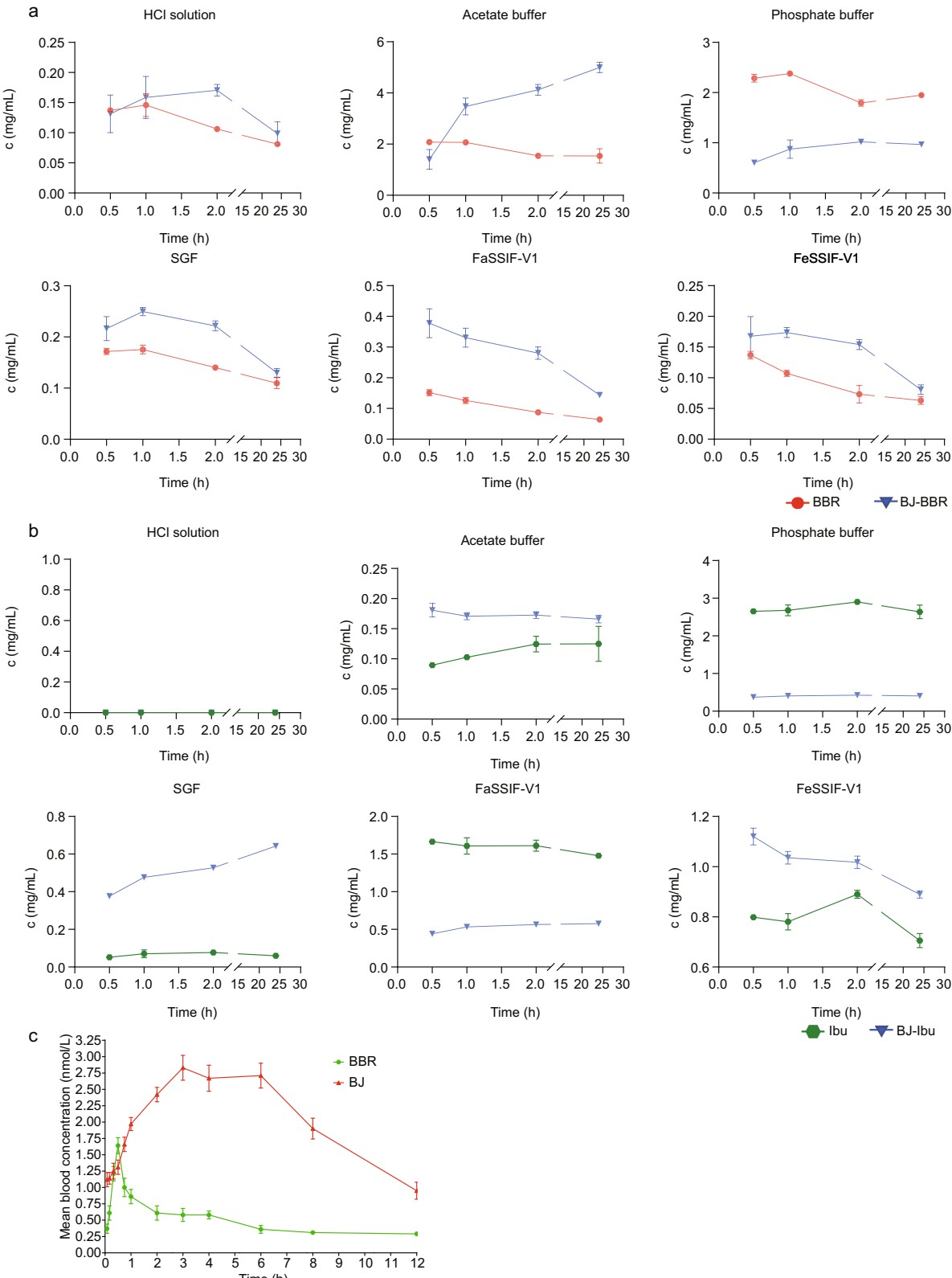

**Fig. 2 BJ exhibits better biofluid solubility and higher bioavailability than BBR. a** Dynamic solubility of BBR (BCl·2H$_2$O) and BBR of BJ in six solutions. **b** Dynamic solubility of Ibu monomer and Ibu of BJ in six solutions. **c** Mean plasma concentration - time curve of BBR (BCl·2H$_2$O) and BBR of BJ in SD rats ($n = 5$ per group).

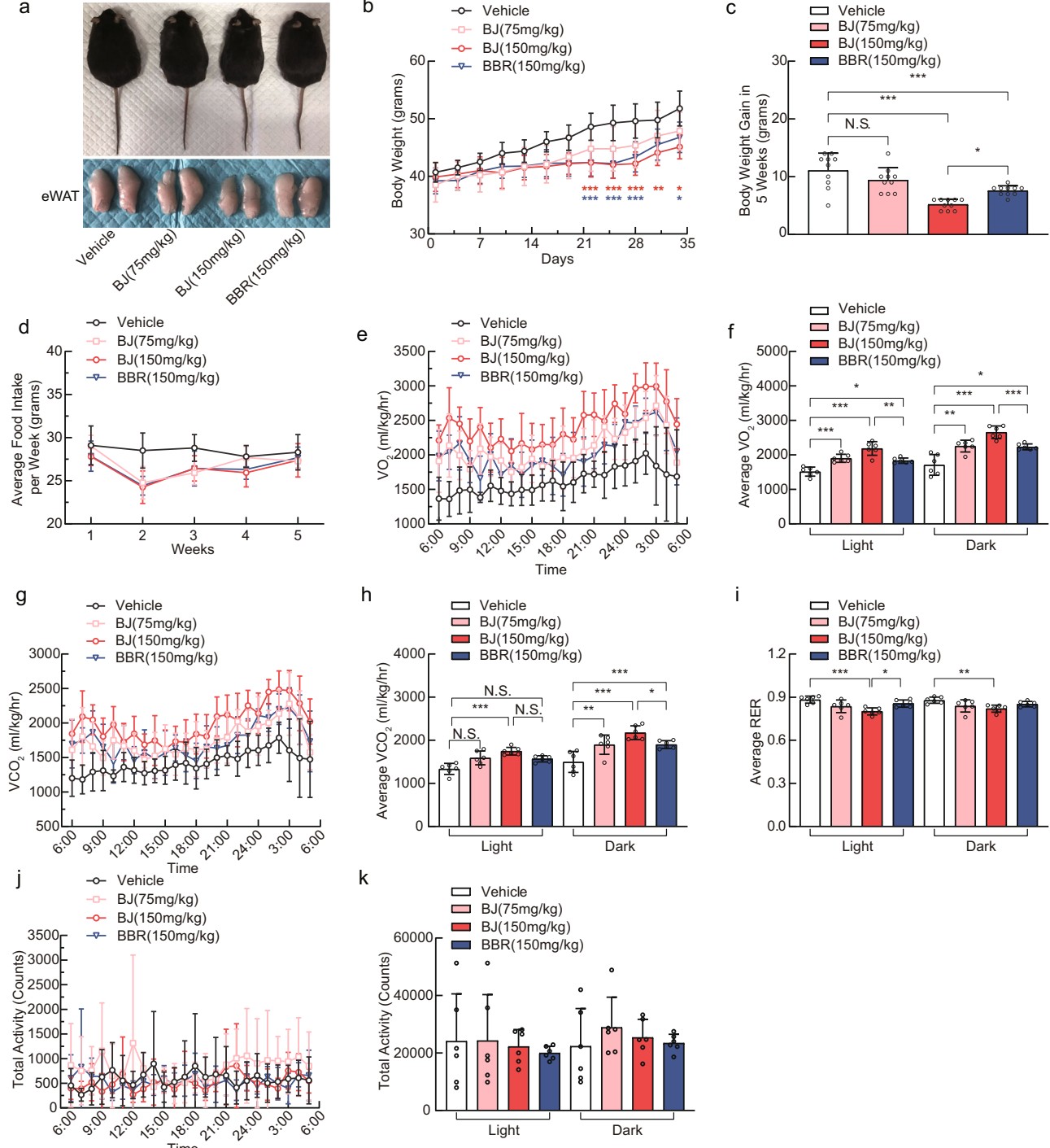

**Fig. 3 BJ reduces body weight and promotes energy expenditure in db/db mice. a** Representative images of db/db mice and their epididymal white adipose tissue (eWAT) form db/db mice treated with BJ or BBR. **b**, **c** Body weight of db/db mice in treatment groups ($n = 10$ per group). **d** Changes of average food intake in each group during 5 weeks of treatment ($n = 6$ per group). **e–h** Oxygen consumption ($VO_2$) (**e**, **f**) and carbon dioxide release ($VCO_2$) (g-h) of db/db mice within 24 h after treatment with BJ or BBR for 5 weeks ($n = 6$ per group). **i** Respiratory exchange ratio ($VCO_2/VO_2$) over a 24-h period in each group ($n = 6$ per group). **j**, **k** Spontaneous horizontal activity of db/db mice over a 24-h period ($n = 6$ per group). Data are expressed as mean ± SD. NS not significant, $^*P < 0.05$, $^{**}P < 0.01$, $^{***}P < 0.001$.

**BJ improves lipid metabolism in db/db mice.** Since BJ significantly reduced the weight of db/db mice, we subsequently assessed the effect of BJ on lipid metabolism in db/db mice. BJ- and BBR -treated db/db mice demonstrated lower fat depots weight than vehicle-treated db/db mice (Fig. 4c, d). HE staining of epididymal adipose tissue (eWAT) form db/db mice showed that

the size of adipocytes in BJ- and BBR-treated db/db mice was significantly reduced compared to that in vehicle-treated db/db mice, and BJ (150 mg/kg)-treated mice had a smaller adipocyte size than the other two treatment groups (Fig. 4a, b). Analysis of lipid levels showed that, the serum levels of total cholesterol (CHO), total triglycerides (TG), free fatty acid (FFA) and

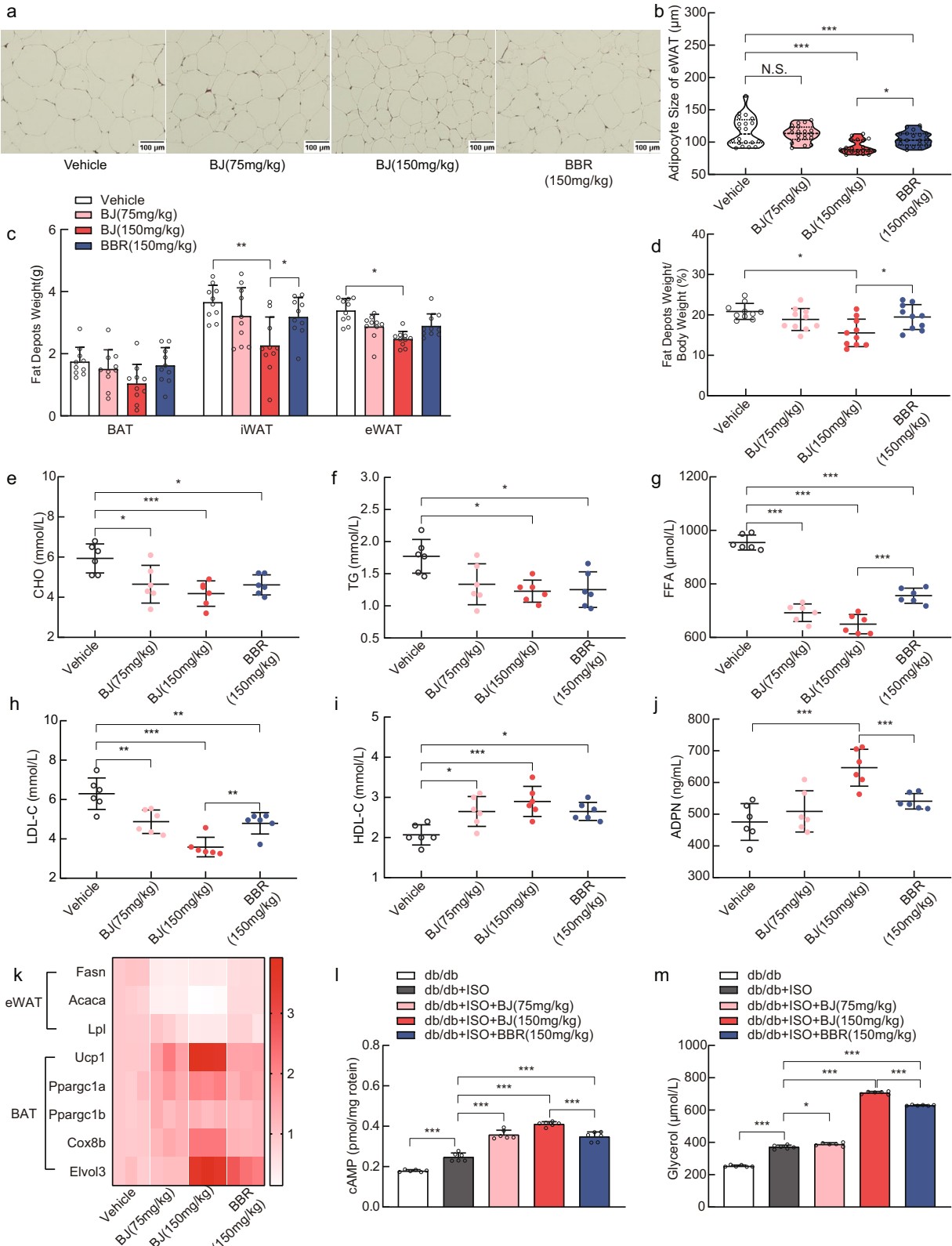

**Fig. 4 BJ regulates lipid metabolism in db/db mice. a**, **b** Representative HE staining images of epididymal adipose tissue (scale bar = 100 μm) and adipocyte size quantification. **c** Weight of BAT, inguinal white adipose tissue (iWAT) and eWAT in different groups of db/db mice (n = 10 per group) (**d**) Percentage of fat depots weight to the body weight (n = 10 per group). **e**–**j** Serum levels of CHO, TG, FFA, LDL-C, HDL-C and ADPN in the fasted state (n = 6 per group). **k** The heat map shows the expression levels of lipid-metabolism-related genes and thermogenic gene. **l**–**m** Measurement of cAMP levels and Glycerol release in eWAT of db/db mice (n = 6 per group). Data are expressed as mean ± SD. NS not significant, *P < 0.05, **P < 0.01, ***P < 0.001.

low-density lipoprotein cholesterol (LDL-C) in db/db mice treated with BJ and BBR were decreased (Fig. 4e–h). In contrast, the serum levels of high-density lipoprotein cholesterol (HDL-C) and adiponectin (ADPN) were increased (Fig. 4i, j). The changes in the BJ (150 mg/kg) treatment group were remarkable. However, the fat depots weight, adipose cell size in eWAT, and serum CHO, TG, FFA, LDL-C, HDL-C and ADPN levels were not significantly different between the vehicle treatment group and Ibu (150 mg/kg) treatment group (Supplementary Fig. 9d–m). In addition to evaluating serum indicators, we also measured the expression of genes related to lipid metabolism in the eWAT of db/db mice. Compared with the vehicle control group, the db/db mice in the BJ and BBR treatment groups exhibited downregulation of lipogenic genes such as fatty acid synthase (Fasn), acetyl CoA carboxylase (Acaca) and lipoprotein lipase (Lpl) in the eWAT (Fig. 4k). Since BAT is an important contributor in energy metabolism, we analyzed the effect of BJ on the expression of thermogenesis-related genes in the BAT of db/db mice. Brown fat enrichment genes, such as Ucp1, Ppargc1a, Ppargc1b, Elovl3, and Cox8b, exhibited elevated express in the BAT of mice in the BJ and BBR treatment groups (Fig. 4k). Moreover, compared with BBR (150 mg/kg) treatment, BJ (150 mg/kg) treatment resulted in a significant effect on lipid metabolism-related gene expression.

Adipose tissue is innervated by the sympathetic nervous system (SNS), and increased SNS activity in adipose tissue can activate β-adrenergic receptors, increase the levels of cAMP, promote thermogenesis and stimulate adipose tissue lipolysis[43–45]. BJ and BBR increased the cAMP levels in the eWAT of db/db mice, promoted the release of glycerol, and enhanced the sensitivity of adipose tissue to isoproterenol stimulation (Fig. 4l, m). BBR (150 mg/kg) treatment exhibited the most significant positive effect. These results showed that BJ improved lipids in db/db mice in a dose-dependent manner, and BJ (150 mg/kg) treatment was more effective than BBR (150 mg/kg) treatment.

**BJ improves glucose metabolism and reduces inflammation of adipose tissue in db/db mice.** To assess the effects of BJ on glucose homeostasis and insulin sensitivity in db/db mice, we monitored fasting glucose levels weekly. Then, we conducted GTT (glucose tolerance test) and ITT (insulin tolerance test) at week 4 and week 5, respectively. The blood glucose levels of the vehicle-treated db/db mice increased from 14.4 ± 3.4 mmol/L at the beginning of the treatment to 23.9 ± 3.4 mmol/L at the end of the treatment. Treatment with BJ (75 mg/kg), BJ (150 mg/kg), and BBR (150 mg/kg) all slowed the increase in glucose levels (13.6 ± 3.0 mmol/L, 9.4 ± 3.6 mmol/L and 11.6 ± 2.4 mmol/L). These findings indicate that all the BBR and BJ treatments improved blood glucose levels and that the effect of BJ (150 mg/kg) treatment was most significant (Fig. 5a). In addition, after 35 days of treatment, BJ (150 mg/kg) significantly reduced the fasting serum insulin levels in the db/db mice, but the BJ (75 mg/kg) or BBR (150 mg/kg) treatments did not produce significant effects (Fig. 5b). Consistent with this finding, glucose and insulin tolerance tests showed that db/db mice were treated with BJ (150 mg/kg) displayed better tolerance to glucose load and were more sensitive to insulin stimulation than the vehicle-, BJ (75 mg/kg) or BBR (150 mg/kg)-treated group (Fig. 5c–f, Supplementary Fig. 10). Of note, serum glucose, insulin levels, and glucose tolerance were not improved in Ibu (150 mg/kg)-treated mice compared with vehicle-treated mice (Supplementary Fig. 9n, s).

Obesity is associated with chronic low-grade inflammation, which often contributes to increased circulating levels of pro-inflammatory cytokines and the development of insulin resistance[9,11]. To evaluate whether BJ could reduce inflammation, we measured the expression levels of inflammatory cytokines in the adipose tissue and serum of db/db mice after 5 weeks of treatment. Compared with vehicle treatment, both BJ and BBR treatment reduced the mRNA levels of tumor necrosis factor-α (TNF-α), interleukin-6 (IL-6), monocyte chemoattractant protein-1 (MCP-1), and the macrophage markers F4/80 (Adgre1),and CD11c (Itgax) in the eWAT of db/db mice (Fig. 5g). Additionally, we found that the serum TNF-α and IL-6 levels of the db/db mice in the BJ- and BBR -treated groups were significantly lower than those in the vehicle-treated group (Fig. 5h, i). Immunohistochemical analysis of epididymal fat pads also showed that the proportion of F4/80-positive macrophages around adipocytes in the BJ- and BBR -treated db/db mice was lower than that of the vehicle-treated db/db mice (Fig. 5j, k). Notably, the above treatment effects of BJ (150 mg/kg) were superior to those of the other two treatment groups.

These data suggest that BJ may improve the glucose metabolism and insulin resistance of db/db mice by reducing the infiltration of macrophages and the release of pro-inflammatory factors in adipose tissue, and it is more effective than the same dose of BBR.

**BJ reduces the expression and activity of IKKε and TBK1 in eWAT of db/db mice.** To understand obesity-related target genes, we used PubMed Abstracts, GeneCards, OMIM, and DisGeNET databases for network analysis, and identified 9481 common obesity-related genes, among which 1791 were screened out due to high relevance scores, such as Tbk1 and Ikbke. To explore the various pathways and physiological functions covered by these obesity-associated genes, we performed pathway analysis using Metascape tools and the KEGG database. The relatively high abundance of KEGG pathways predicted by Metascape were insulin resistance, cAMP, PPAR, and Apelin pathways, in which insulin resistance, PPAR signaling pathway, and Apelin signaling pathway contain AMPK/UCP1 pathway (Fig. 6a). In addition, studies have shown that the non-canonical IκB kinase (IKK) plays a crucial role in obesity and metabolic diseases[13–15]. To investigate the underlying mechanism by which BJ treatment inhibits obesity and improves metabolism, we analyzed the activity and expression levels of IKKε and TBK1 in adipose tissue. The results indicated that the mRNA levels of IKKε and TBK1 in the eWAT of db/db mice were significantly increased compared with those in the eWAT of WT mice from the same background (Fig. 6d, e). Moreover, the immunocomplex kinase assay and western blotting analysis showed that the expression levels and activity of IKKε and TBK1 were increased in db/db eWAT (Fig. 6b, c, g, h). Both doses of BJ treatment significantly reduced the mRNA and protein levels of IKKε and TBK1, as well as the activity of these two kinases (Fig. 6b–h). Subsequently, phosphorylation of AMPK, a direct downstream target of TBK1, was measured.

We found that AMPKα Thr172 phosphorylation (p-AMPKα) was inhibited in the eWAT of db/db mice and was restored by BJ treatment (Fig. 6k). BJ treatment also increased the expression of uncoupling protein 1 (UCP1) and the phosphorylation of TBK1 at Ser172 (Fig. 6i, j). The increase in TBK1 phosphorylation may be an indirect result of AMPK activation[15,46].

Considering the importance of BAT as a thermogenic organ in regulating thermogenesis and energy balance[44], we assessed the effect of BJ on IKKε and TBK1 in BAT of db/db mice. IKKε and TBK1 kinase assay results showed that neither dose of BJ affected the expression levels of IKKε and TBK1 in BAT (Fig. 6l, m). Analysis by real-time PCR produced comparable results (Fig. 6n, o). This may be related to the low-level of inflammation in BAT. These data indicated that BJ reduced the activity and expression levels of IKKε and TBK1 in WAT of db/db mice but had no effect on the activity and expression levels of these two kinases in BAT.

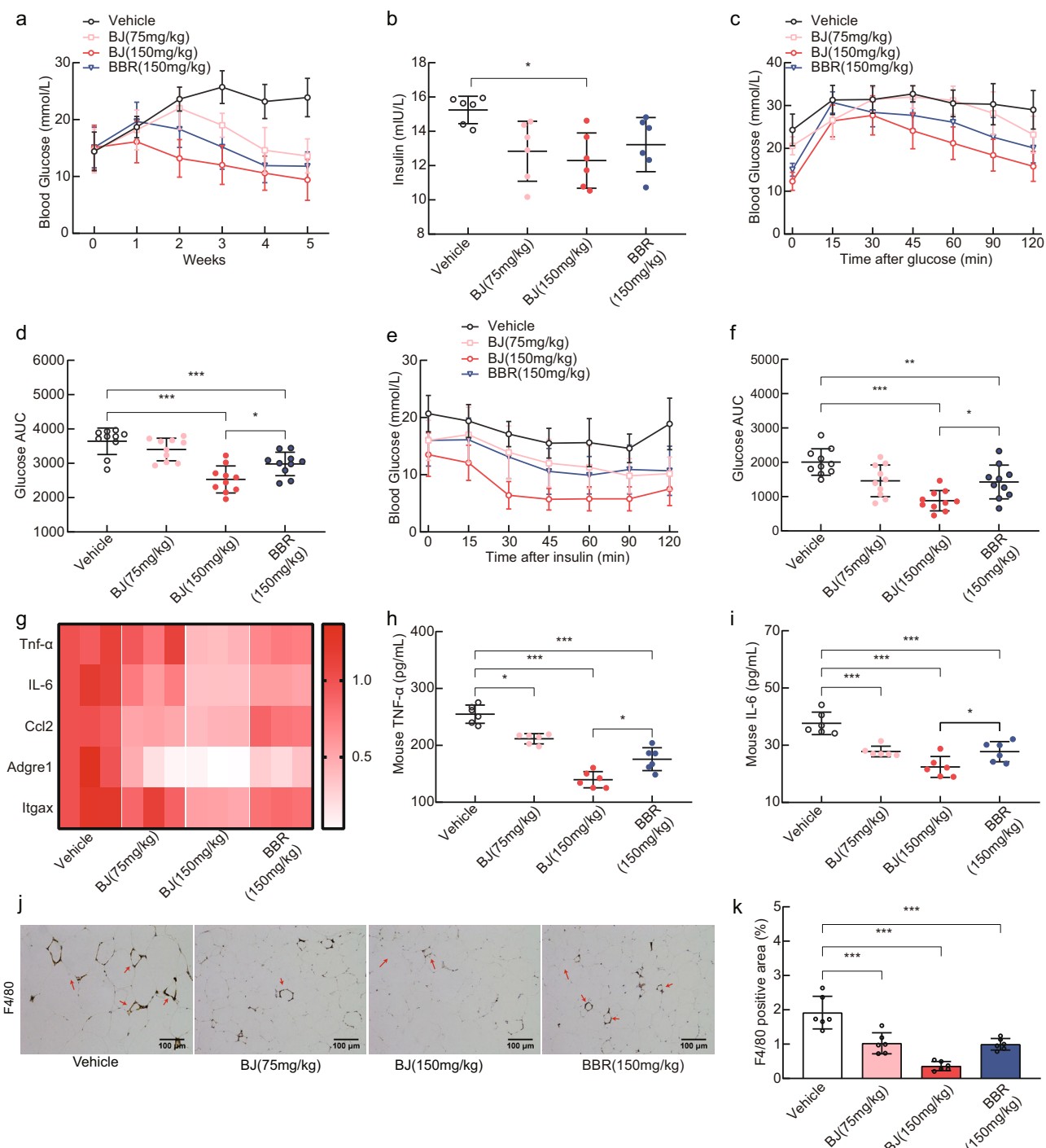

**Fig. 5 BJ regulates blood glucose levels, enhances insulin sensitivity and inhibits inflammation of eWAT in db/db mice. a** Weekly fasting glucose levels of db/db mice ($n = 6$ per group). **b** Fasting serum insulin levels in db/db mice after 35 days of treatment ($n = 6$ per group). **c**, **d** GTT and the area under the curve (AUC) ($n = 6$ per group). **e**, **f** ITT and the AUC ($n = 6$ per group). **g** Heat map of inflammatory related genes and macrophage marker genes in eWAT of db/db mice. **h**, **i** Serum levels of TNF-α and IL-6 in db/db mice ($n = 6$ per group). **j** Immunohistochemical staining of the eWAT for F4/80 (scale bar = 100 μm). **k** Quantification of immunohistochemical staining as a percentage of positive area ($n = 6$). Data are expressed as mean ± SD. NS not significant, *$P < 0.05$, **$P < 0.01$, ***$P < 0.001$.

**BJ inhibits lipid accumulation and inflammation in 3T3-L1 adipocytes**. Meanwhile, we conducted in vitro experiments to validate the role of BJ in adipocytes. BJ (3 μM and 10 μM) or BBR (10 μM) was added on day 8 during the differentiation of 3T3-L1 adipocytes and incubated for 24 h to evaluate the inhibitory effect of BJ on lipid accumulation. Compared with control differentiated adipocytes, BJ reduced the accumulation of lipid droplets in 3T3-L1

adipocytes in a dose-dependent manner (Fig. 7a, b). Staining with MitoTracker red and an anti-UCP1 antibody indicated that the mitochondrial mass of 3T3-L1 adipocytes treated with BJ was more significant, and the expression of the UCP1 protein was significantly increased (Fig. 7c, d). Furthermore, compared with the same dose of BBR, the effect of BJ was more significant. These findings were similar to the results in vivo.

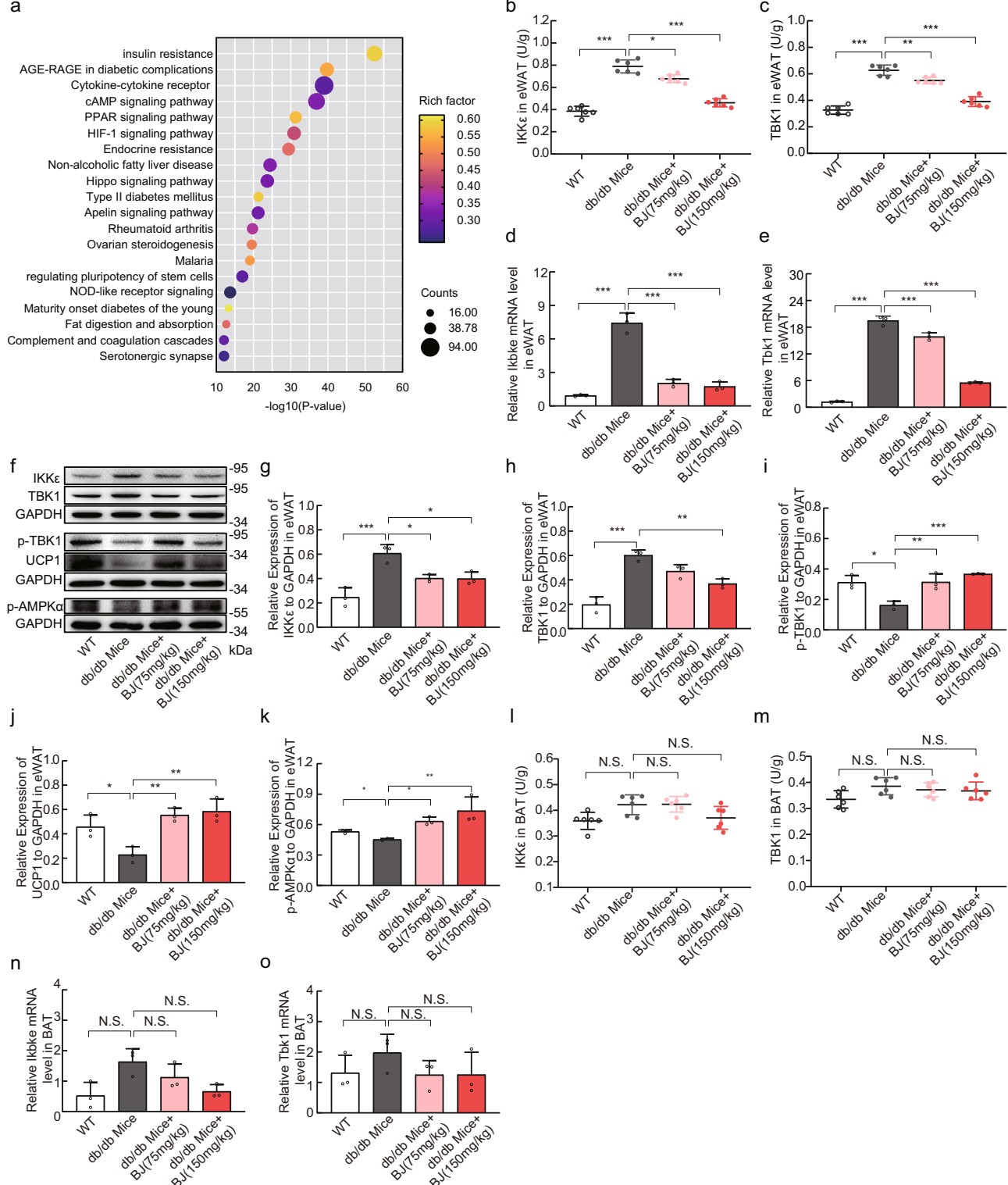

**Fig. 6 BJ inhibits the expression of IKKε and TBK1 in eWAT of db/db mice, but has no effect on BAT. a** Top 20 enriched pathways of the KEGG database after identification of the genes associated with obesity. **b**, **c** IKKε and TBK1 kinase activity in the eWAT of WT and db/db mice ($n = 6$ per group). **d**, **e** RT-qPCR analysis of the relative expression levels of Ikbke and Tbk1 in eWAT of WT and db/db mice ($n = 3$ per group). **f**–**k** Relative protein levels of IKKε, TBK1, p-TBK1, UCP1, and p-AMPKα in eWAT of WT and db/db mice by western blotting ($n = 3$ per group). **l**, **m** IKKε and TBK1 kinase activity in the BAT of WT and db/db mice ($n = 6$ per group). **n**, **o** RT-qPCR analysis for mRNA expression levels of Ikbke and Tbk1 in BAT ($n = 3$ per group). Data are expressed as mean ± SD. NS not significant, *$P < 0.05$, **$P < 0.01$, ***$P < 0.001$.

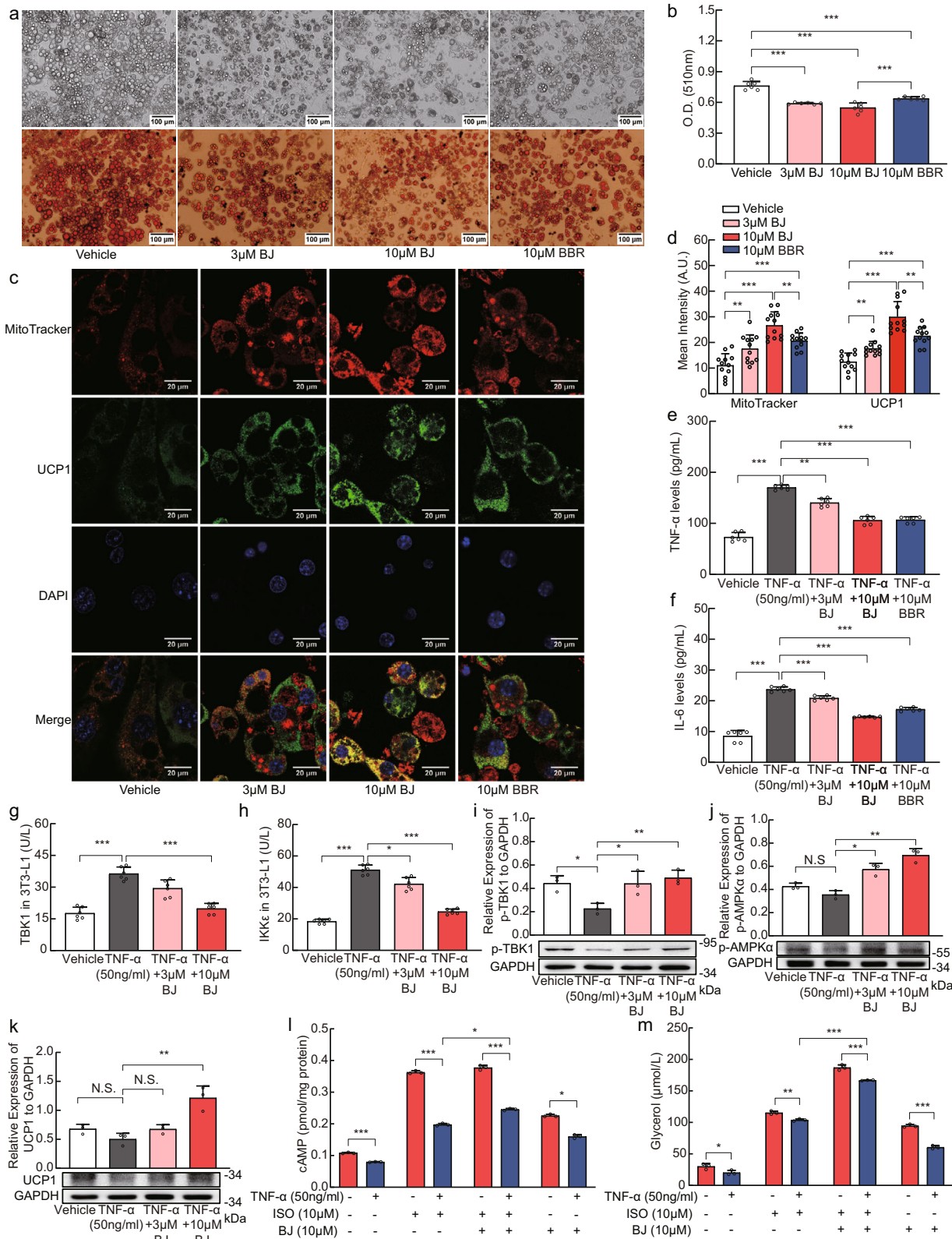

We next examined the role of BJ on TNF-α induced inflammation of differentiated 3T3-L1 adipocytes. The results showed that the levels of TNF-α and IL-6 were significantly increased in the supernatant of 3T3-L1 adipocytes treated with TNF-α (Fig. 7e, f). Supplementation with BJ and BBR significantly reduced the levels of TNF-α and IL-6, but there was no significant difference in the effect of between BJ and BBR treatments of the same dose.

Since TNF-α has been shown to stimulate the expression of IKKε and TBK1 in 3T3-L1 adipocytes[13], we investigated the effect of BJ on the expression of IKKε and TBK1 in TNF-α induced 3T3-L1 adipocyte inflammation. The expression levels of IKKε and TBK1 were significantly increased after 48 h of TNF-α induction (Fig. 7g, h). In addition, the expression levels of p-AMPKα, UCP1, and p-TBK1 decreased, albeit with no statistical

**Fig. 7 BJ improves lipid accumulation and energy metabolism of 3T3-L1 adipocytes and inhibits inflammation of 3T3-L1 adipocytes. a, b** Representative *Oil Red O* staining of differentiated 3T3-L1 adipocytes (scale bar = 100 μm) and spectrophotometric analysis. **c** Immuno-fluorescence staining of UCP1 (green), MitoTracker (red), and DAPI (blue) in differentiated 3T3-L1 adipocytes (scale bar = 20 μm). **d** Quantification of fluorescence intensity of MitoTracker and UCP1 ($n = 12$ per group). **e, f** Effects of BJ on the levels of TNF-α and IL-6 in supernatant of differentiated 3T3-L1 adipocytes induced by TNF-α ($n = 6$ per group) (**g, h**) Effects of BJ on the TBK1 and IKKε kinase activity in differentiated 3T3-L1 adipocytes induced by TNF-α ($n = 6$ per group). **i–k** Relative protein levels of p-TBK1, p-AMPKα and UCP1 in differentiated 3T3-L1 adipocytes induced by TNF-α ($n = 3$ per group). **l, m** Measurement of cAMP levels and Glycerol release in differentiated 3T3-L1 adipocytes induced by TNF-α ($n = 3$ per group). Data are expressed as mean ± SD. NS not significant, *$P < 0.05$, **$P < 0.01$, ***$P < 0.001$.

significance (Fig. 7i, j, k). After BJ treatment for 24 h, the expression levels of IKKε and TBK1 markedly decreased, and the expression of p-AMPKα, UCP1, and p-TBK1 increased significantly (Fig. 7g, h, i, j, k). In addition, we pretreated 3T3-L1 adipocytes with BJ and stimulated differentiated 3T3-L1 adipocytes with isoproterenol. Consistent with our in vivo findings, BJ promoted glycerol release, increased cAMP levels, and blocked the inhibitory effect of TNF-α on isoproterenol-stimulated adrenergic signaling (Fig. 7l, m). These data indicated that BJ can also inhibit lipid accumulation, reduce adipocyte inflammation, and improve adipocyte sensitivity to catecholamine in vitro.

**BJ regulates TBK1/AMPK/UCP1 pathway and IKKε/adrenergic/cAMP pathway in 3T3-L1 adipocytes**. Since BJ inhibited the expression of TBK1 and IKKε in vivo and in vitro, we overexpressed TBK1 or IKKε in 3T3-L1 adipocytes to evaluate whether BJ directly inhibits TBK1 or IKKε and examine its regulatory role in adipocyte energy metabolism. As shown in Fig. 8a, Supplementary Fig. 11a, c, there was no difference between the blank control group (BC) and the empty vector group (empty vector served as a negative control, NC). Compared with the empty vector group, TBK1 protein levels were significantly increased in the TBK1 transfection group, which confirmed successful overexpression of TBK1 in 3T3-L1 adipocytes (Fig. 8a, Supplementary Fig. 11a, c). Previous studies have shown that AMPK is a direct downstream substrate of TBK1[15], and TBK1 overexpression significantly reduces AMPKα Thr172 phosphorylation and the expression of UCP1 (Fig. 8c–g, Supplementary Fig. 11b, d, f and h). Compared with TBK1 transfection, BJ treatment markedly reduced the TBK1 protein levels, increased the phosphorylation of AMPKα Thr172 and increased the UCP1 protein levels (Fig. 8c–g, Supplementary Fig. 11b, d, f and h). Consistent with the above analysis results, BJ also increased the phosphorylation of TBK1 at Ser172 (Fig. 8b, Supplementary Fig. 11e, g). This result may be due to the mitigation of feedback inhibition.

In addition, western blotting analysis also confirmed the expression of IKKε in 3T3-L1 adipocytes. The IKKε protein level was significantly decreased and the UCP1 protein level was significantly increased after 24 h of BJ treatment (Fig. 8h, i). This observation suggests that BJ can regulate UCP1 transcription by directly inhibiting IKKε. It has been shown that IKKε inhibits mitochondrial biogenesis by weakening the adrenergic/cAMP pathway, while TBK1 does not play a role in controlling the sensitivity of adipose tissue to catecholamine[15,18]. We then stimulated differentiated 3T3-L1 adipocytes with isoproterenol (ISO) to further explore the role of BJ in regulating adipocyte susceptibility to catecholamine. Compared with the empty vector control, ISO treatment induced significant increases in UCP1 protein expression level, glycerol release, and the production of the second messenger cAMP (Fig. 8j–l). The inhibition of ISO efficacy by IKKε overexpression was ameliorated by treating transfected 3T3-L1 adipocytes with 10 μM BJ (Fig. 8j–l). This is consistent with previous observations of TNFα-induced adipocyte inflammation.

These data further suggest that BJ can directly inhibit TBK1 to regulate AMPK-mediated mitochondrial biogenesis in adipocytes, and it also directly inhibits IKKε, increase adipocyte sensitivity to catecholamine, promote fat decomposition, and increase energy consumption.

**BJ targets TBK1 and IKKε to regulate 3T3-L1 adipocytes**. To further determine the role of TBK1 and IKKε kinases in BJ regulation of adipocytes, we knocked down the expression of TBK1 or IKKε in 3T3-L1 adipocytes with siRNA and treated the adipocytes with BJ for 24 h. As shown in Fig. 9a, b, TBK1 expression was successfully inhibited by the transfection of specific siRNA. Additionally, phosphorylation of AMPKα Thr172 and UCP1 expression were significantly increased (Fig. 9a, c–g). BJ alone significantly increased the expression of p-AMPKα and UCP1, and promoted mitochondrial biogenesis in 3T3-L1 adipocytes. Compared with TBK1-knockdown, the expression levels of p-AMPKα and UCP1 were not significantly changed in TBK1-knockdown 3T3-L1 adipocytes treated with BJ (Fig. 9a, c-g).

Western blotting confirmed the successful knockdown of IKKε by transfection with siRNA (Fig. 9h). IKKε knockdown in 3T3-L1 adipocytes simultaneously increased UCP1 expression (Fig. 9i). Similarly, the expression of UCP1 was significantly increased in 3T3-L1 adipocytes treated with BJ alone (Fig. 9i). Compared with IKKε-knockdown, UCP1 expression was not significantly altered in IKKε-knockdown 3T3-L1 adipocytes treated with BJ (Fig. 9i). Furthermore, IKKε knockdown or BJ treatment alone significantly increased cAMP accumulation, glycerol release, and UCP1 expression in ISO-stimulated 3T3-L1 adipocytes (Fig. 9j–l). However, compared with IKKε-knockdown, BJ did not significantly increase the above changes in IKKε-knockdown 3T3-L1 adipocytes in response to ISO stimulation (Fig. 9j–l). Taken together, our data suggests that the lack of additional effect of BJ in the context of TBK1 or IKKε knockdown provides evidence that BJ is likely exerting its effects through these kinases.

**Discussion**

In this paper, BJ was prepared by the solution reaction method. Through a series of characterizations, we found that if the reaction medium contains water, berberine may combine two molecules of water to form hydrogen bonds between berberine and ibuprofen. BJ also shows improvements in solubility and bioavailability compared with BBR.

Moreover, we explored the effects of BJ on body weight and metabolism in db/db mice based on previous pharmacological studies of BBR. We demonstrated that db/db mice treated with BJ exhibited significant weight loss, which may be associated with increased oxygen consumption, leading to increased energy turnover. BJ also improved glucose and lipid homeostasis, insulin resistance, and reduced adipose tissue inflammation in db/db mice. It is noteworthy that although BJ is a co-crystal formed by hydrogen bonding between BBR and Ibu, Ibu did not affect the body weight, fat depots weight, serum lipid contents, serum glucose and insulin levels, and glucose tolerance. This suggests

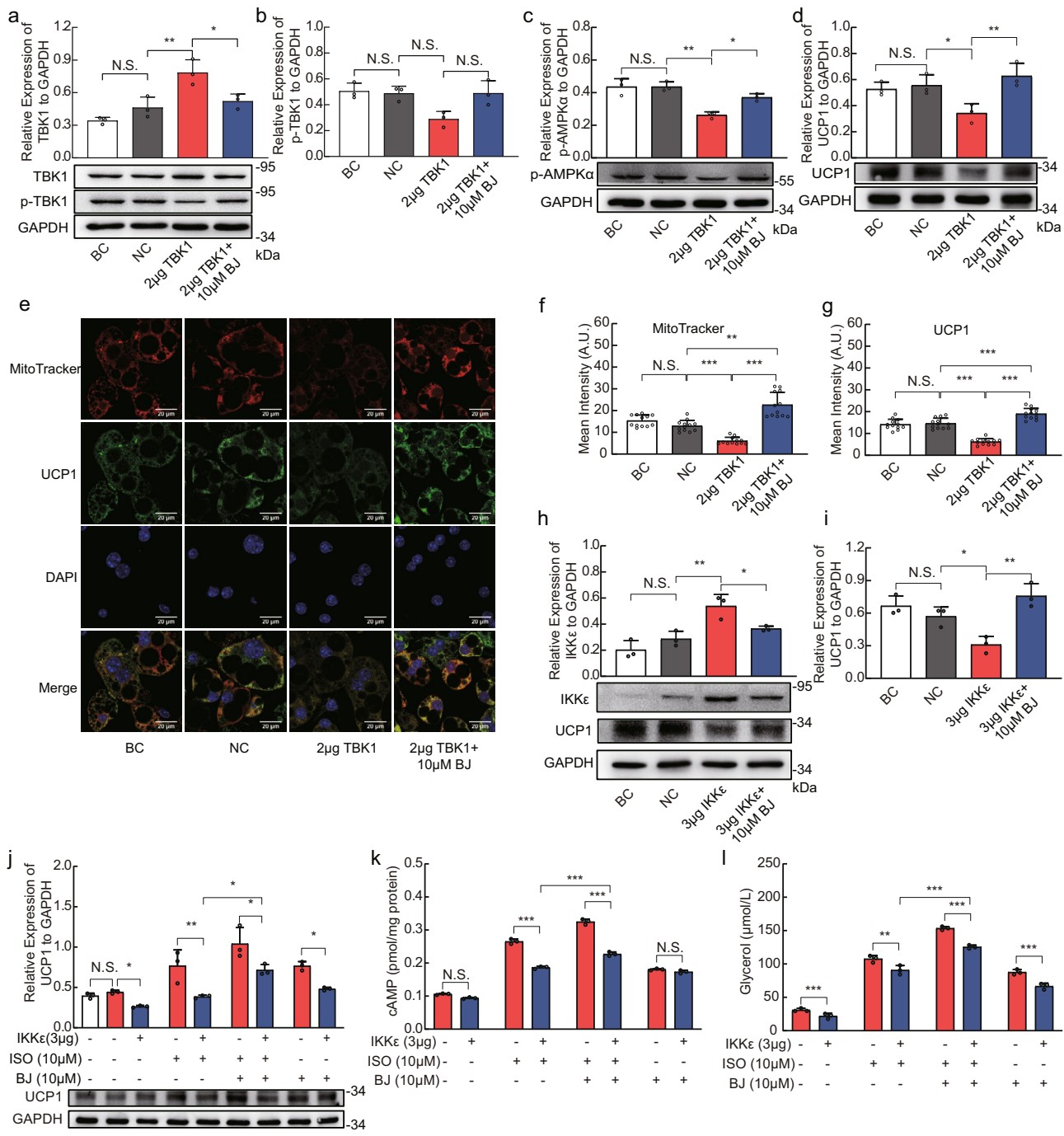

**Fig. 8 BJ regulates TBK1 / AMPK / UCP1 pathway and IKKε / adrenergic / cAMP pathway in 3T3-L1 adipocytes. a–d** Effect of BJ on relative protein levels of TBK1, p-TBK1, p-AMPKα and UCP1 in differentiation of 3T3-L1 adipocytes transfected with TBK1 plasmid or empty vector. **e** Immunofluorescence staining of UCP1 (green), MitoTracker (red), and DAPI (blue) in differentiated 3T3-L1 adipocytes transfected with TBK1 plasmid or empty vector followed by treatment with or without BJ (scale bar = 20 μm). **f, g** Quantification of fluorescence intensity of MitoTracker and UCP1 (n = 12 per group). **h, i** Effect of BJ on relative protein levels of IKKε (**h**) and UCP1 (**i**) in differentiation of 3T3-L1 adipocytes transfected with IKKε plasmid or empty vector. **j** Relative protein levels of UCP1 in differentiated 3T3-L1 adipocytes transfected with ISO or IKKε plasmid or empty vector. **k, l** Measurement of cAMP levels (**k**) and Glycerol release (**l**) in differentiated 3T3-L1 adipocytes transfected with ISO or IKKε plasmid or empty vector. Data are expressed as mean ± SD. NS not significant, *$P < 0.05$, **$P < 0.01$, ***$P < 0.001$.

that Ibu alone has no potential effect on obesity. In vitro studies, different doses of BJ reduced lipid accumulation and TNF-α induced inflammation in 3T3-L1 adipocytes and promoted adipocyte energy metabolism. We also compared the metabolism of BJ and BBR and found that BJ and BBR produced comparable results in the same experiment. Nevertheless, the effect of BJ was

better, which may be related to the improvement of bioavailability.

Recent studies revealed that chronic inflammation plays a leading role in the pathogenesis of obesity[47–50]. The pro-inflammatory factors (such as TNF-α, IL- 6, and IL-1β) secreted by M1-macrophages in adipose tissue act on the receptors in

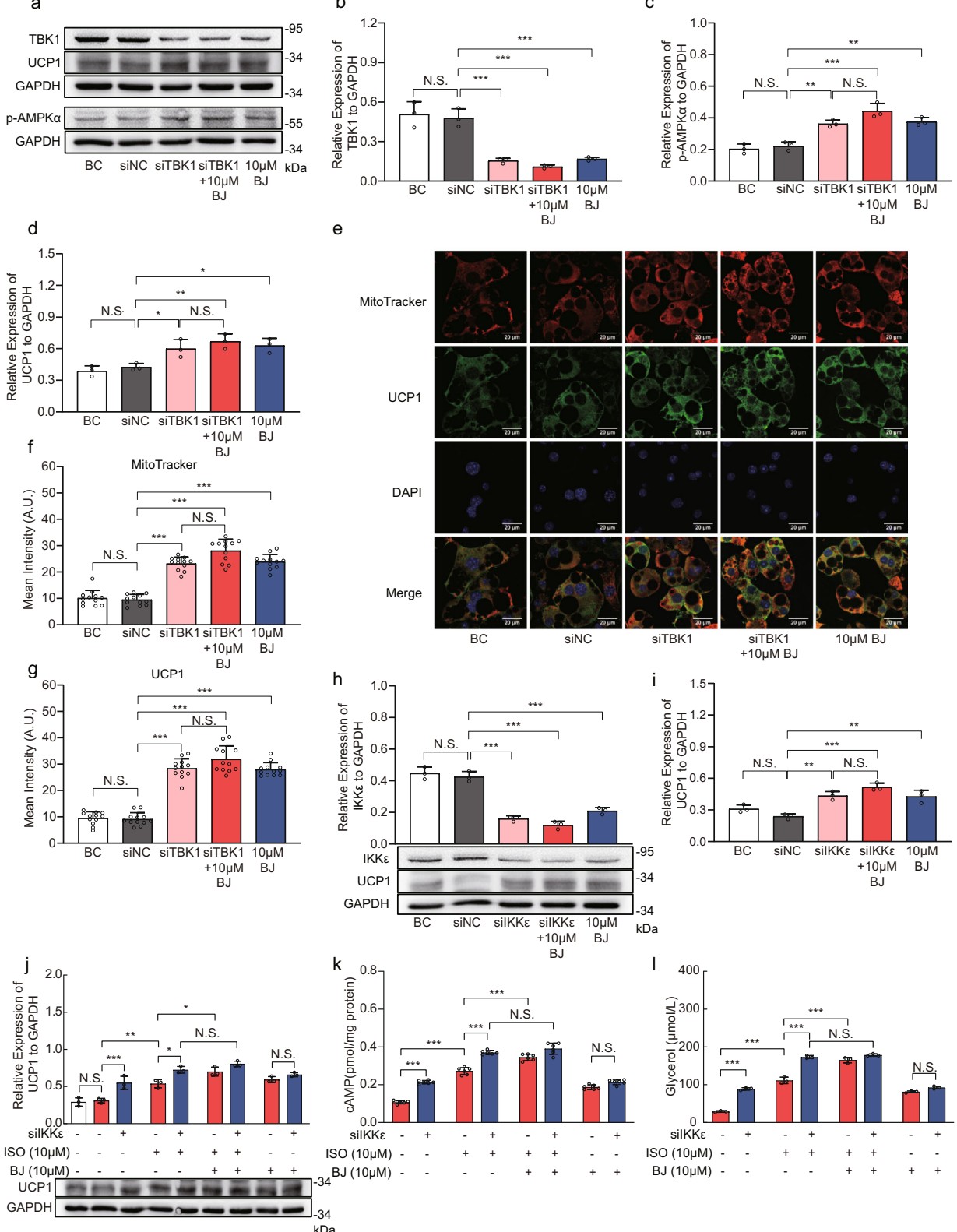

**Fig. 9 BJ targets TBK1 and IKKε to regulate 3T3-L1 adipocytes. a–d** Relative protein levels of TBK1, p-AMPKα and UCP1 in TBK1-knockdown 3T3-L1 adipocytes treated with or without BJ. **e–g** Immunofluorescence staining of UCP1 (green), MitoTracker (red), and DAPI (blue) in TBK1-knockdown 3T3-L1 adipocytes treated with or without BJ (scale bar = 20 μm). **h, i** Relative protein levels of IKKε and UCP1 in IKKε-knockdown 3T3-L1 adipocytes treated with or without BJ. **j** Relative protein levels of UCP1 in IKKε-knockdown 3T3-L1 adipocytes treated with ISO or BJ. **k, l** Measurement of cAMP levels (**k**) and Glycerol release (**l**) in IKKε-knockdown 3T3-L1 adipocytes treated with ISO or BJ. Data are expressed as mean ± SD. NS not significant, *$P < 0.05$, **$P < 0.01$, ***$P < 0.001$.

hypertrophic adipocytes, thus inducing the secretion of pro-inflammatory cytokines (especially MCP1 (Ccl2)) by adipocytes, this process can promote the transformation of anti-inflammatory M2-macrophages into inflammatory M1-macrophages and the infiltration of M1-macrophages into adipocytes[51,52]. Increased inflammation and infiltration of macrophages in adipose tissue inhibit insulin-stimulated glucose uptake, leading to insulin resistance[53–55]. In vivo, we observed that the levels of inflammatory cytokines TNF-α, IL-6, and MCP1 were significantly down-regulated in db/db mice treated with BJ. Measurements of the expression of macrophage markers, such as F4/80 (Adgre1) and CD11c (Itgax) in adipose tissue and the staining of F4/80 in adipose tissue showed that BJ could reduce the infiltration of macrophages into adipose tissue. Our in-vitro experiments in 3T3-L1 adipocytes showed that BJ could inhibit TNF-α induced adipocyte inflammation. These results suggest that improvements in insulin sensitivity after BJ treatment may be associated with reductions in adipose tissue inflammation.

A positive energy balance leads to obesity, which is associated with a low-grade inflammatory state involving activation of the NF-κB pathway and enhanced expression of TBK1 and IKKε[13]. In this study, we observed that BJ significantly reduced the levels of TBK1 and IKKε in TNF-α induced inflammatory models of 3T3-L1 adipocytes and in the eWAT of db/db mice. However, BJ had little effect on the expression and activity of TBK1 and IKKε in BAT, which may be related to the low level of inflammation in BAT. TBK1 directly inhibits the phosphorylation of AMPKα at *Thr172* site (p-AMPKα) in adipocytes[15], which is a crucial sensor of the cellular energy state. Phosphorylated AMPKα-*Thr172* regulates UCP1 and mitochondrial biogenesis. Interestingly, p-AMPKα, UCP1, and p-TBK1 were significantly up-regulated by BJ treatment in the eWAT of db/db mice and in 3T3-L1 adipocytes, and the increased levels of p-TBK1 may be due to indirect stimulation of TBK1 by AMPK activation. Thus, it is hypothesized that BJ may regulate AMPK phosphorylation by inhibiting the expression of TBK1, which in turn regulates metabolism.

In the obese state, WAT is resistant to catecholamines, which reduces the production of cAMP induced by β-adrenoceptor activation, leading to the reduction of lipolysis and lipid oxidation, as well as the expression of mitochondrial bioregulatory transcription factors[56]. In the present study, we found that BJ increased the sensitivity of adipocytes to isoproterenol stimulation, promoted the accumulation of cAMP and the release of glycerol, and increased the expression of UCP1 in vivo and in vitro. Since IKKε reduces the sensitivity of the β-adrenergic receptor to catecholamine in adipocytes of obese mice, the content of cAMP is decreased, suggesting that BJ may reduce lipid accumulation and promote energy metabolism by inhibiting the IKKε/adrenergic /cAMP pathway.

To further investigate the mechanism by which BJ inhibits obesity, we overexpressed or knocked down TBK1 and IKKε in 3T3-L1 adipocytes in vitro. The results showed that BJ could directly inhibit the expression of TBK1 and IKKε, and BJ had no additional effect in the context of TBK1 or IKKε knockdown, which further supports our above hypothesis. Since TBK1 and IKKε are activated by obesity-related inflammation and BJ directly inhibits IKKε and TBK1, the reduction in inflammation observed after BJ treatment in this study may be partiallly and indirectly related due to the indirect effect of improving to the alleviation of metabolic disease.

In summary, our study not only developed the co-crystal BJ to address the limitation of low bioavailability of BBR, but also found that BJ treatment can improve obesity. The mechanism by which BJ reduces obesity may be partly due to its inhibitory effects on the expression of TBK1 and IKKε, and may occur through two different pathways: the TBK1/AMPK/UCP1 pathway and the IKKε/adrenergic/cAMP pathway. These results suggest that the non-canonical IκB kinases TBK1 and IKKε are potential therapeutic targets in adipocytes to prevent obesity.

## Methods

**Materials**. Berberine chloride dihydrate (Berberine hydrochloride, purity: 98%) was obtained from Yuanye Inc. (Shanghai, China) and Ibuprofen (purity: 98%) was obtained from Aladdin (Shanghai, China). Both of them were used directly after purchase. All other solvents were analytically pure. Deionized water was used in this study. All plasmids were supplied by Shanghai GenePharma Co. Ltd (Shanghai, China). Sources of antibodies were as follows: IKKε antibody (Cell SignalingTechnology, 2690), TBK1 antibody (Cell Signaling Technology, 3013), Phospho-TBK1 *Ser172* antibody (Cell Signaling Technology, 5483), Phospho-AMPKα *Thr172* antibody (Cell Signaling Technology, 2535), Anti-UCP1 antibody (Abcam, ab10983), F4/80 antibody (proteintech, 28463-1-AP), GAPDH antibody (proteintech, 60004-1-Ig), Anti-rabbit IgG (Cell Signaling Technology, 7074 S), Anti-mouse IgG (Cell Signaling Technology, 7076 S), CL488-conjugated Affinipure Goat Anti-Mouse lgG (H + L) (proteintech, SA00013-1), CL488-conjugated Affinipure Goat Anti-Rabbit lgG (H + L) (proteintech, SA00013-2). 3-Isobutyl-1-methylxanthine (IBMX), dexamethasone (DEX), insulin, ISO and Oil Red O solution (0.5% in isopropanol) were purchased from Sigma-Aldrich (St Louis, MO, USA). Recombinant Murine TNF-α was purchased from Pepro Tech (Rocky Hill, NJ, USA). The BCA protein assay kit was purchased from Beyotime Biotechnology Co., Ltd (Beijing, China).

**Animals**. Male pathogen-free SD rats, 250 ± 10 g, were purchased from the Experimental Animal Center in QingLongShan (Nanjing, China). Obese and diabetic C57BLKS/J-Leprdb/Leprdb (db/db) male mice (average age, 6 weeks) and age-matched C57BLKS/Nju (wildtype, WT) male mice were purchased from Jiangsu Jicui Yaokang Biotechnology Co., Ltd (Nanjing, China). Prior to the experiments, the animals were all housed with the standard feeding conditions: with temperature of 25.0 ± 1.0 °C, humidity of 60 ± 10%, and 12 h light/dark cycles. The mice were provided free access to a standard diet and water. All animal procedures have been approved by the Animal Care and Use Committee of Nanjing University of Chinese Medicine.

**Cell culture and differentiation of 3T3-L1 preadipocytes**. Murine 3T3-L1 pre-adipocytes were the kind gift of Prof. Wenbin Shang and Prof. Xizhong Yu (Nanjing University of Chinese Medicine, China). Cells were cultured in Dulbecco's modified Eagle's medium (DMEM) supplemented with 10% (vol/vol) fetal bovine serum (FBS), 1% (vol/vol) penicillin/streptomycin in an incubator at 37 °C and 5% CO₂ (until confluent). Two days after the preadipocytes reached 100% confluency (designated day 0), the cells were cultured in DMEM with 10% FBS supplemented with IBMX (0.5 mM final concentration), DEX (1 μM final concentration), and insulin (10 μg/ml final concentration) for 3 days (days 0–3). Then the cells were cultured in a medium containing insulin (10 μg/ml final concentration) for another 3 days (days 3–6). The medium was then changed to DMEM with 10% FBS and replaced every 2 days until more than 90% of cells showed adipocyte morphology.

**Preparation of the BJ**. NaOH (200 mg, 5 mmol) and ibuprofen (1031.5 mg, 5 mmol) were placed in a three-necked flask. The deionized water (25 mL) was added dropwise and the solution was stirred thoroughly. Then BBR (BBR chloride dihydrate, 1.86 g, 5 mmol) and 25 ml of ethyl acetate were added. After reaction completely, the ethyl acetate layer was separated and then cooled down to obtain the sediment.

**Nuclear magnetic resonance (NMR)**. Take 5 mg of sample respectively, dissolve them with 0.6 ml of deuterated methanol (Methanol-d4) reagent, and conduct ¹H NMR analysis.

**Thermal gravimetric analysis (TGA) and differential scanning calorimetry (DSC)**. TGA analysis was conducted under a nitrogen atmosphere using instrument (Discovery 5500 or Q5000). The sample was put on an aluminum pan and heated to 300 °C with a 10 °C/min heating rate and a 25 mL/min nitrogen flow. DSC experiment was carried out in a nitrogen atmosphere using instrument (TA Discovery 2500 or Q2000). DSC curve was also recorded, the sample was heated to 250 °C at a 10 °C/min heating rate.

**Powder X-ray diffraction (PXRD)**. PXRD pattern was obtained from a LYNXEYE_XE_T(1D mode) diffractometer using a Cu/Kα (λ = 1.5406 Å, 40 kV and 40 mA) radiation. Data were collected in the range from 3° to 40° with a scan rate of 0.02°. The sample was pressed on a silicon holder prior to data collection.

**Single crystal X-ray diffraction (SCXRD)**. The crystal structure was determined by SCXRD. The sample was obtained by the solvent evaporation technique. In brief, a certain amount of the product was dissolved in 10 mL ethanol and stirred

for about 15 min. Afterwards, the solution was filtered through paper filters, allowing the solvent to slowly evaporate. The diffraction data was collected with a Bruker Smart APEX II diffractometer (Mo-Kα radiation, λ = 0.71073 Å) at 296(2) K, and multi-scan absorption correction was conducted by using SADABS. Data integration and reduction was performed by using the SHELXL-2017 software package.

**Dynamic vapor sorption (DVS)**. A certain mass of the compound is weighed and the ambient temperature inside the apparatus is set to 25 °C. Cycle: 40-0-95-0-40% RH. Stage Step: 10%. Equilibrium: 0.002 dm/dt (%/min). Minimum dm/dt stability duration: 60 min. Maximum dm/dt stage time: 360 min

**In vitro dissolution studies**. Solubilities were determined by adding excess sample to 30 mL different mediums, such as HCl solution, acetate buffer, phosphate buffer, SGF, FaSSIF and FeSSIF. The configuration of various buffer solutions is shown in Supplementary Table 2. After being shaken at 400 rpm at 37 °C for 0.5, 1, 2, and 24 h, 0.3 mL of suspension was sampled and centrifuged at 14,000 rpm for 8 min to isolate mother liquor, and then analyzed by HPLC after dilution. All samples were analyzed in triplicate.

**In vivo pharmacokinetics studies**. Ten male pathogen-free SD rats were randomly and averagely divided into two groups, refraining from food (except water) 12 h before the study. The two groups were respectively intragastrically given with BBR suspension (dispersed in sodium carboxymethyl cellulose (0.5% CMC-Na) solution) and BJ suspension (dispersed in 0.5% CMC-Na solution), at the dose of 115 mg/kg. Blood samples (0.5 mL) were collected from the retro-orbital venous plexus at appropriate time points and placed in collection tubes with anticoagulants. Plasma was obtained from the blood samples by centrifuged (4000 rpm, 10 min, 4 °C) and stored at −20 °C until analysis. 400 μL of acetonitrile was added to 200 μL of rat plasmas, and vortexed for 90 s and finally centrifuged at 12,000 rpm for about 10 min. 20 μL of the supernatant was used to determinate the concentration of berberine in plasma by using the liquid chromatography-mass spectrometry (Waters XBridge, Waters, USA).

The maximum peak concentration ($C_{max}$) and the time of reaching $C_{max}$ ($T_{max}$) can be obtained from the measured data. The area under the concentration-time curve ($AUC_{0 \rightarrow t}$) was estimated by the trapezoidal summation. The area under the curve from time zero to infinity ($AUC_{0 \rightarrow \infty}$) was also determined.. Lastly, the relative bioavailability was calculated by the formula: $F_{rel} = F = AUC_{(0-\infty) BJ} / AUC_{(0-\infty) BBR} * 100\%$.

**The effects and potential mechanism by which BJ improves obesity**. At 7 weeks of age, Obese and diabetic C57BLKS/J-Leprdb/Leprdb (db/db) male mice and age-matched C57BLKS/Nju (wildtype, WT) male mice were randomly divided into five groups, each containing at least six mice: lean wild type control mice, db/db control mice, db/db mice treated with BJ (75 mg/kg), db/db mice treated with BJ (150 mg/kg), db/db mice treated with BBR (150 mg/kg), db/db mice treated with Ibu (150 mg/kg). BBR, BJ or Ibu suspension in 0.5% CMC-Na solution was given by gavage once per day for 35 days. The lean wild type control and the db/db control mice were given equal vehicle volume (0.5% CMC-Na solution). All mice were monitored for food intake, body weight, and fasting blood glucose during the 35-day trial. Glucose tolerance test (GTT) and insulin resistance test (ITT) were performed at 4 and 5 weeks of treatment, respectively. Metabolic analysis was performed at 5 weeks of treatment. At the end of the study, blood samples were taken through the retroorbital venous sinus of mice, fat depots (brown fat, inguinal subcutaneous fat, epididymal fat, perirenal fat and mesenteric fat depot) were dissected, weighed, and immediately frozen in liquid nitrogen and stored at −80 °C until required. Weight of fat depots are expressed in percentage (fat depots weight /bodyweight * 100%).

**Glucose tolerance test (GTT) and Insulin tolerance test (ITT)**. After being fasted for 12 h, the mice were given an intraperitoneal injection of glucose (1.2 g/kg) for a glucose tolerance test. Mice were fasted for 4 h for insulin resistance test and intraperitoneally injected with 1 unit/kg of insulin. Blood glucose levels were measured from the tail vein blood at 0, 15, 30, 45, 60, 90, and 120 min after injection. Plasma insulin levels were measured using an ultrasensitive mouse insulin ELISA kit (Nanjing Jiancheng Bioengineering Institute, Nanjing, China).

**Indirect calorimetry and motor activity**. At week 5 of treatment, basal metabolic parameters of mice were measured using the Comprehensive Lab Animal Monitoring System (CLAMS, Columbus Instruments, Columbus, OH). The mice were adapted to the metabolic cage for 24 h, and food intake, oxygen consumption ($VO_2$), and carbon dioxide release ($VCO_2$) were measured over the next 48 h. The number of times the infrared beams were broken along the X- or Y-axis was used to measure the dynamic activity of mice. The respiratory exchange rate (RER) was calculated using classical Lusk theory and equations $RER = VCO_2/VO_2$.

**Histology and immunohistochemistry**. The adipose tissue was fixed overnight in 4% paraformaldehyde, then embedded in paraffin and cut into 5 μm thick slices.

Tissue sections were stained with hematoxylin and eosin (H&E). For immunohistochemistry, sections were incubated with primary antibodies against F4/80 (1:100 dilution) diluted in the blocking solution at 4 °C overnight. The tissue sections were then incubated with the horseradish peroxidase (HRP)-conjugated secondary antibody at room temperature for 2 h. Finally, the sections were developed with diaminobenzidine (DAB) and counterstained with hematoxylin. Images were obtained under a microscope. To quantify the F4/80-positive macrophages, two random fields per sample were analyzed by Image J software. The IHC profiler plugin of ImageJ was used to calculate the F4/80 positive area.

**Cell transfection**. 3T3-L1 adipocytes were transfected on day 8 of differentiation to overexpress TBK1 or IKKε. According to the manufacturer's instructions, differentiated 3T3-L1 adipocytes were transfected with 2 μg TBK1 expression plasmid, 3 μg IKKε expression plasmid, or the corresponding empty vector using Lipofectamine 2000 (Invitrogen, Carlsbad, CA, USA) for 6 h. Small interference RNA transfection (siRNA) was performed using 4 μM TBK1 siRNA (siTBK1) or 4 μM IKKε siRNA (siIKKε) or corresponding NC siRNA (siNC).

**Oil Red O Staining and Lipid Quantification**. Lipid accumulation in differentiated adipocytes was observed by *Oil Red O* staining. The differentiated 3T3-L1 adipocytes were washed with PBS for 3 times and fixed with 4% paraformaldehyde for 30 min at room temperature. The fixed 3T3-L1 adipocytes were washed with PBS for 3 times and stained with *Oil Red O* solution at room temperature for 60 min. Stained lipid droplets were photographed and then eluted with 100% isopropanol stained *Oil Red O*. Finally, the optical density (OD) value of the eluent was measured at 510 nm wavelength with a microplate reader.

**Immunofluorescence staining**. The 3T3-L1 adipocytes were washed with PBS for 3 times, fixed with 4% paraformaldehyde solution for 30 min, and then washed again for 3 times. The fixed samples were permeated with 0.2% Triton X-100 for 20 min and sealed in 5% normal goat serum in PBS for 1 h. The cells were then incubated overnight at 4 °C with the appropriate primary antibodies (1:100 dilution). The next day, the 3T3-L1 adipocytes were washed and incubated at room temperature with goat anti-rabbit secondary antibody or goat anti-mouse secondary antibody (1:1000 dilution) for 2 h and washed with PBS for 3 times. The nuclei were stained with DAPI. For mitochondrial detection, the 3T3-L1 adipocytes were incubated with 100 nM MitoTracker Red CMXRos (Solarbio, Beijing, China) in the dark at 37 °C for 30 min. Fluorescence images were taken using a laser confocal microscope (SP8, Leica, Germany). At least four images per section were acquired for quantification, and the mean intensity were evaluated using ImageJ.

**Biochemical analyses**. The serum was collected by centrifugation at $3000 \times g$ at 4 °C for 15 min and stored at −80 °C until analysis. TG, TC, LDL-C, HDL-C, and FFA were routinely tested on a Chemray 240 automatic biochemical analyzer (Chemray 240, Shenzhen, China). Serum levels of mouse adiponectin, interleukin6 (IL-6), tumor necrosis factor-α (TNF-α) were measured using commercial enzyme-linked immunosorbent assay (ELISA) kits (Nanjing Jiancheng Bioengineering Institute, Nanjing, China) and according to the manufacturer's instructions. The levels of IL-6, TNF-α in 3T3-L1 adipocyte culture supernatant were also detected by ELISA.

**Measurement of glycerol and cAMP content**. The epididymal adipose tissue was dissected, the fresh adipose tissue was weighed and rapidly rinsed with PBS. The adipose tissue was cut and incubated with isoproterenol (10 μM) for 2 h, and homogenized in a cold cleavage buffer to prepare the adipose tissue homogenate. Then the homogenate was centrifuged at $12,000 \times g$ at 4 °C for 15 min, and the supernatant was collected. For 3T3-L1 adipocytes, the adipocytes were treated with or without 50 ng/ml TNF-α or 3 μg IKKε plasmid or 4 μM siIKKε for 24 h and then treated with or without 10 μM ISO or 10 μM BJ for 24 h. Then the cells were lysed in a petri dish, scraped off, and centrifuged at $12,000 \times g$ at 4 °C for 15 min before the supernatant was collected.

Glycerol or cAMP content in adipose tissue and 3T3-L1 adipocytes was determined by a glycerol assay kit (Nanjing Jiancheng Bioengineering Institute, Nanjing, China) or a mouse cAMP assay kit (Nanjing Jiancheng Bioengineering Institute, Nanjing, China). And concentrations were further standardized according to protein concentration.

**IKKε or TBK1 activity assays**. The adipose tissue was ultrasound-crushed in PBS buffer solution (1:9 w/v) to prepare the tissue homogenate. The tissue homogenate was centrifuged at $5000 \times g$ for 10 min, and the supernatant was taken for detection. Briefly, samples or standards were added to the microplates pre-coated with IKKε antibody or TBK1 antibody. Then horseradish peroxidase (HRP) labeled antibody was added and incubated at 37 °C for 60 min. After thorough washing, the 3,3′,5,5′-tetramethylbenzidine (TMB) substrate was used for color development, and the absorbance (OD value) was determined at 450 nm.

**Quantitative real-time polymerase chain reaction analysis (qRT-PCR)**. Total RNA was isolated from 3T3-L1 cells and adipose tissue homogenates using Trizol

reagent (Invitrogen, Carlsbad, CA, USA). RNA concentration and purity were examined at 260/280 nm using a nanodrop spectrophotometer (NanoDrop One, Thermo Fisher Scientific, USA). The total RNA was reverse transcribed to cDNA using the EasyScript All-in-One First-Strand cDNA Synthesis SuperMix for qPCR kit (TransGen Biotech, Beijing, China). The quantitative reverse transcription polymerase chain reaction was performed using TransStart Top Green qPCR SuperMix (TransGen Biotech, Beijing, China) and an Applied Biosystems 7500 RealTime PCR System (Life Technologies, USA). Relative expression levels were normalized to gapdh expression and quantified using the $2^{-\Delta\Delta Ct}$ method. The primers were synthesized by Shanghai Sangon Biotech, and the detailed primer sequences were shown in Supplementary Table 5 in the supplementary materials.

**Western blot analysis**. The adipose tissue or 3T3-L1 adipocytes were homogenized by ultrasound in an ice-cold RIPA lysis buffer containing 1% PMSF and centrifuged at $12,000 \times g$ at 4°C for 15 min to collect the supernatant. The BCA protein assay kit (Beyotime, Shanghai, China) was used to detect the total protein concentration. Proteins (20-30 μg) of the sample were separated by 10% SDS-PAGE gel and transferred onto polyvinylidene difluoride (PVDF) membranes. Then the PVDF membranes were blocked in 5% skim milk powder in TBS containing 1% Tween-20 (TBST) for 2 h at room temperature and washed with TBST for 3 times. The membranes were incubated with the appropriate primary antibody, IKKε (1:1000 dilution), TBK1 (1:1000 dilution), p-TBK1 *Ser172* (1:1000 dilution), p-AMPKα *Thr172* (1:1000 dilution), UCP1 (1:1000 dilution), and GAPDH (1:1000 dilution) in the blocking buffer solution at 4°C overnight. After washing, the membranes were incubated with horseradish peroxidase-conjugated secondary antibodies (in 5% skim milk powder) for 1 h at room temperature. Finally, the membranes were treated with enhanced chemiluminescence agents (Tanon, Shanghai, China) and the signals were monitored using the Tanon 5200 multi chemiluminescent imaging system (Tanon, Shanghai, China).

**Statistics and reproducibility**. All experiments were repeated at least three times. All data were obtained from at least three independent experiments and were expressed as mean ± SD. Statistical analysis was conducted using GraphPad Prism V.8.00. Comparisons of means in two groups were made using an unpaired t-test. Ordinary one-way ANOVA was used for intergroup analysis, followed by Tukey post-hoc analysis. Normality and homogeneity of variances were tested. When data did not comply with the homogeneity of variances, Brown Forsythe & Welch one-way ANOVA with Dunnett's multiple comparisons test was performed for comparison of multiple groups. A $P < 0.05$ was considered statistically significant.

**Reporting summary**. Further information on research design is available in the Nature Research Reporting Summary linked to this article.

## Data availability

Crystallographic data of BJ are deposited in the Cambridge Crystallographic data center (CCDC) database under the accession code CCDC - 2183784. Source data of the figures in this paper can be found in Supplementary Data 1. Uncropped blots of Figs. 6f, 7i–k, 8a, c, d, h, j, 9a, h and j are provided as Supplementary Figs. 12a–e and 13a–c. Any additional information required to reanalyze the data reported in this paper is available from the lead contact upon request.

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

## Acknowledgements
The authors thank Prof. Wenbin Shang and Prof. Xizhong Yu for the kind gift of the murine 3T3-L1 preadipocyte cell line. This work was supported by Key Project of Jiangsu Province for Fundamental Research and Development (BE2018717), Nanjing High-level Entrepreneur Talent Introduction Program (2019, Prof. Jing Zhu), Doctor of Entrepreneurship and Innovation in Jiangsu Province (2021, JSSCBS20210364) and Natural Science Foundation of Jiangsu Province-Youth Fund (BK20210684).

## Author contributions
These authors contributed equally: Man Wang, Rong Xu, Xiaoli Liu. J.Z. designed this project. M.W. and R.X. performed the experiments and analyzed the data. M.W., R.X. and X.L. contributed to the writing of the manuscript. M.W., R.X., X.L., L.Z., S.Q., Y.L., P.Z., M.Y. and J.Z. critically revised the manuscript.

## Competing interests
The authors declare no competing interests.
