## [Peer Review File · Communications Biology]

Reviewers' comments:

Reviewer #1 (Remarks to the Author):

In this study, the authors engineered a drug co-crystal (BJ) by combining berberine (BBR) with ibuprofen. Compared to BBR, BJ had better oral bioavailability and superior anti-obesity effects in db/db mice compared to BBR. The authors conducted several experiments to investigate the molecular mechanisms through which BJ acts to improve metabolic health and show evidence that the TBK1 and IKKe pathways are involved.

This article would be of interest to researchers in several fields, such as pharmacology, metabolism, and physiology. The novel co-crystal is an innovative approach to improve the utilisation of a compound that has previously shown promising effects in animal models. This study comprises an impressive quantity of work, with a good range of both ex vivo and in vivo experiments to support the conclusions. However, the attention to detail and the quality of the manuscript could be improved.

Specific comments:

1. The authors do not discuss the possibility of additional effects of the ibuprofen beyond changes in oral bioavailability and solubility. This is particularly relevant when in vitro studies show different effects of BJ and BBR at the same dose.

2. The study shows overexpression experiments, but not any gene knockdown. It could be informative to perform experiments involving TBK1 and IKKe knockdown. If the effects of BJ/BBR are blunted by lack of TBK1/IKKe, this would be stronger evidence to suggest that BJ/BBR act through these kinases.

3. Lines 50-51. Authors claim that clinical studies have shown upregulation of IKKe and TBK1 in patients with NAFLD by citing work by Oral et al. (reference 17). However, the paper cited did not measure IKKe and TBK1 in human patients, but rather tested an IKKe/TBK1 inhibitor in obese patients. While the abstract states that TBK1 and IKKe are increased with obesity, the introduction explains that this was previously shown in mice, not humans. The authors should correct the wording of their statement.

4. Lines 91-92. Reference 26 seems to be out of place as the cited paper does not appear to include human subjects. Perhaps it should be reference 27 instead?

5. As the methods section is at the end of the paper, abbreviations for the different buffer solutions (SGF, FaSSIF-V1, FeSSIF-V1) should be defined in the results section where they first appear (line 168).

6. Figure 4d shows "Fat Mass (%)". The figure legend states that this is the "weight of adipose tissue" but it is not clear what this actually is. Is it the sum of the adipose tissue weights at termination divided by the body mass? If so, it must include tissues other than BAT, iWAT and eWAT because the numbers in Fig 4c do not appear to add up. Or is it a non-invasive measurement of total body fat mass such as DEXA or EchoMRI? If so, this should be described in the methods and should not be called "weight of adipose tissue" since it would also include lipid content from other tissues.

7. Figure 5f. The ITT data are quantified by measuring the area under the curve. But these changes are exaggerated due to differences in fasting glucose. The initial slope of the curve (0-15 min) would be a better indicator of insulin sensitivity.

8. "Vehicle" is spelt incorrectly in Figure 5g.

9. In Figure 5j, authors show only four representative images of F4/80 staining and claim that the proportion of F4/80-positive macrophages around adipocytes was decreased in BJ- and BBR-treated mice. The authors should support their argument with quantification of several random fields of view from each section for each mouse. The methods section (4.4.12) also does not state how the F4/80-positive macrophages were counted or quantified. In fact, the authors have stated that they used biotinylated secondaries but no streptavidin-HRP or substrate, and that they performed DAPI staining on IHC samples then observed with a fluorescence microscope (lines 685-688). This section should be checked and corrected.

10. Figure 7a-c. BJ appears to have stronger effects than BBR in vitro when both are administered at 10 μ M. It could be informative to discuss possible reasons for this phenomenon since it cannot be due to a difference in oral bioavailability (see comment 1).

11. Figure 7c. The authors claim that UCP1 expression is increased in BJ-treated 3T3-L1 cells

compared to both vehicle and BBR, but there are only a few representative IF images shown. Was any fluorescence intensity quantification performed on these data? Also, MitoTracker is spelt incorrectly.

12. Lines 505-508. It is not clear what "the latter" is referring to. I initially thought "the latter" was referring to 3T3-L1 adipocytes, but after multiple reads, I realised the intended meaning is that p-TBK1 upregulation may be due to feedback from AMPK activity. The sentence should be rewritten for clarity.

13. Line 514. "Adrenaline" should be replaced with "isoproterenol" for accuracy.

14. Line 515. The authors did not show changes in mitochondrial biogenesis in vivo.

15. Authors state that the One-Way ANOVA was performed to assess changes between groups. However, they do not state whether normality or homogeneity of variance was tested. If the data fail the normality test by a meaningful amount, a non-parametric test should be used (or the data should be transformed e.g., log-transformed to achieve normality). If homogeneity of variance is not true, a Welch's ANOVA should be performed instead.

16. Overall, the quality of the writing should be improved.

Reviewer #2 (Remarks to the Author):

This paper "A co-crystal berberine-ibuprofen improves obesity by inhibiting the protein kinases TBK1 and IKK ϵ " studied the pharmacology effect of co-crystal BJ on obesity and its related metabolic diseases. It is very impressive that the effects on lipid metabolism, glucose metabolism, and inflammation in db/db mice were superior to berberine. However, the manuscript would be significantly improved by the following:

1. This article wanted to describe the effect of the co-crystal BJ, but it only compares with the effect of berberine alone, so I wonder whether ibuprofen has some effects on regulating obesity. How to exclude the influence of ibuprofen? We hope that the authors can make a more convincing statement in introduction, or give direct experimental pieces of evidence in results.
2. Since BJ could promote the oral absorption of BBR and significantly enhance the bioavailability of BBR, what's the accumulation of BJ in adipose tissue compared with BBR alone?
3. Why the wild type group was missing when evaluate the effect of BJ on body weight, energy expenditure, lipid, and glucose metabolism.
4. In line 86, "oral BBR (1000 mg/d or 15000 mg/d) could reduce body weight, inflammation...", 15000 mg/d of BBR is far more than the usual oral dosage. Please check it.
5. It is confused that BJ could both decrease the expression of TBK1 at mRNA level and inhibit the phosphorylation of TBK1 protein, since they couldn't occur simultaneously.
6. In Figure 8e and 8f, the results of Immunofluorescence staining for P-TBK1 and AMPK were lost.
7. The manuscript requires extensive editing for English grammar and sentence structure.

Reviewers' comments:

Reviewer #1 (Remarks to the Author):

In this study, the authors engineered a drug co-crystal (BJ) by combining berberine (BBR) with ibuprofen. Compared to BBR, BJ had better oral bioavailability and superior anti-obesity effects in db/db mice compared to BBR. The authors conducted several experiments to investigate the molecular mechanisms through which BJ acts to improve metabolic health and show evidence that the TBK1 and IKKe pathways are involved.

This article would be of interest to researchers in several fields, such as pharmacology, metabolism, and physiology. The novel co-crystal is an innovative approach to improve the utilisation of a compound that has previously shown promising effects in animal models. This study comprises an impressive quantity of work, with a good range of both ex vivo and in vivo experiments to support the conclusions. However, the attention to detail and the quality of the manuscript could be improved.

Specific comments:

1. The authors do not discuss the possibility of additional effects of the ibuprofen beyond changes in oral bioavailability and solubility. This is particularly relevant when in vitro studies show different effects of BJ and BBR at the same dose.
2. The study shows overexpression experiments, but not any gene knockdown. It could be informative to perform experiments involving TBK1 and IKKe knockdown. If the effects of BJ/BBR are blunted by lack of TBK1/IKKe, this would be stronger evidence to suggest that BJ/BBR act through these kinases.
3. Lines 50-51. Authors claim that clinical studies have shown upregulation of IKKe and TBK1 in patients with NAFLD by citing work by Oral et al. (reference 17). However, the paper cited did not measure IKKe and TBK1 in human patients, but rather tested an IKKe/TBK1 inhibitor in obese patients. While the abstract states that TBK1 and IKKe are increased with obesity, the introduction explains that this was previously shown in mice, not humans. The authors should correct the wording of their statement.
4. Lines 91-92. Reference 26 seems to be out of place as the cited paper does not appear

to include human subjects. Perhaps it should be reference 27 instead?

5. As the methods section is at the end of the paper, abbreviations for the different buffer solutions (SGF, FaSSIF-V1, FeSSIF-V1) should be defined in the results section where they first appear (line 168).

6. Figure 4d shows “Fat Mass (%)”. The figure legend states that this is the “weight of adipose tissue” but it is not clear what this actually is. Is it the sum of the adipose tissue weights at termination divided by the body mass? If so, it must include tissues other than BAT, iWAT and eWAT because the numbers in Fig 4c do not appear to add up. Or is it a non-invasive measurement of total body fat mass such as DEXA or EchoMRI? If so, this should be described in the methods and should not be called “weight of adipose tissue” since it would also include lipid content from other tissues.

7. Figure 5f. The ITT data are quantified by measuring the area under the curve. But these changes are exaggerated due to differences in fasting glucose. The initial slope of the curve (0-15 min) would be a better indicator of insulin sensitivity.

8. “Vehicle” is spelt incorrectly in Figure 5g.

9. In Figure 5j, authors show only four representative images of F4/80 staining and claim that the proportion of F4/80-positive macrophages around adipocytes was decreased in BJ- and BBR-treated mice. The authors should support their argument with quantification of several random fields of view from each section for each mouse. The methods section (4.4.12) also does not state how the F4/80-positive macrophages were counted or quantified. In fact, the authors have stated that they used biotinylated secondaries but no streptavidin-HRP or substrate, and that they performed DAPI staining on IHC samples then observed with a fluorescence microscope (lines 685-688). This section should be checked and corrected.

10. Figure 7a-c. BJ appears to have stronger effects than BBR in vitro when both are administered at 10 uM. It could be informative to discuss possible reasons for this phenomenon since it cannot be due to a difference in oral bioavailability (see comment 1).

11. Figure 7c. The authors claim that UCP1 expression is increased in BJ-treated 3T3-L1 cells compared to both vehicle and BBR, but there are only a few representative IF

images shown. Was any fluorescence intensity quantification performed on these data? Also, MitoTracker is spelt incorrectly.

12. Lines 505-508. It is not clear what “the latter” is referring to. I initially thought “the latter” was referring to 3T3-L1 adipocytes, but after multiple reads, I realised the intended meaning is that p-TBK1 upregulation may be due to feedback from AMPK activity. The sentence should be rewritten for clarity.

13. Line 514. “Adrenaline” should be replaced with “isoproterenol” for accuracy.

14. Line 515. The authors did not show changes in mitochondrial biogenesis in vivo.

15. Authors state that the One-Way ANOVA was performed to assess changes between groups. However, they do not state whether normality or homogeneity of variance was tested. If the data fail the normality test by a meaningful amount, a non-parametric test should be used (or the data should be transformed e.g., log-transformed to achieve normality). If homogeneity of variance is not true, a Welch’s ANOVA should be performed instead.

16. Overall, the quality of the writing should be improved.

Reviewer #2 (Remarks to the Author):

This paper “A co-crystal berberine-ibuprofen improves obesity by inhibiting the protein kinases TBK1 and IKK ϵ ” studied the pharmacology effect of co-crystal BJ on obesity and its related metabolic diseases. It is very impressive that the effects on lipid metabolism, glucose metabolism, and inflammation in db/db mice were superior to berberine. However, the manuscript would be significantly improved by the following:

1. This article wanted to describe the effect of the co-crystal BJ, but it only compares with the effect of berberine alone, so I wonder whether ibuprofen has some effects on regulating obesity. How to exclude the influence of ibuprofen? We hope that the authors can make a more convincing statement in introduction, or give direct experimental pieces of evidence in results.

2. Since BJ could promote the oral absorption of BBR and significantly enhance the bioavailability of BBR, what’s the accumulation of BJ in adipose tissue compared with BBR alone?

3. Why the wild type group was missing when evaluate the effect of BJ on body weight, energy expenditure, lipid, and glucose metabolism.
4. In line 86, “oral BBR (1000 mg/d or 15000 mg/d) could reduce body weight, inflammation...”, 15000 mg/d of BBR is far more than the usual oral dosage. Please check it.
5. It is confused that BJ could both decrease the expression of TBK1 at mRNA level and inhibit the phosphorylation of TBK1 protein, since they couldn't occur simultaneously.
6. In Figure 8e and 8f, the results of Immunofluorescence staining for P-TBK1 and AMPK were lost.
7. The manuscript requires extensive editing for English grammar and sentence structure.

Point-by-Point response

Reviewer #1 (Remarks to the Author):

In this study, the authors engineered a drug co-crystal (BJ) by combining berberine (BBR) with ibuprofen. Compared to BBR, BJ had better oral bioavailability and superior anti-obesity effects in db/db mice compared to BBR. The authors conducted several experiments to investigate the molecular mechanisms through which BJ acts to improve metabolic health and show evidence that the TBK1 and IKKe pathways are involved.

This article would be of interest to researchers in several fields, such as pharmacology, metabolism, and physiology. The novel co-crystal is an innovative approach to improve the utilisation of a compound that has previously shown promising effects in animal models. This study comprises an impressive quantity of work, with a good range of both ex vivo and in vivo experiments to support the conclusions. However, the attention to detail and the quality of the manuscript could be improved.

1. The authors do not discuss the possibility of additional effects of the ibuprofen

beyond changes in oral bioavailability and solubility. This is particularly relevant when *in vitro* studies show different effects of BJ and BBR at the same dose.

Response: We thank the Reviewer for pointing out this issue. To explore the potential impact of Ibu on obesity, we treated db/db mice with Ibu (150 mg/kg) or vehicle for 5 weeks. Ibu treatment did not affect body weight and fat depots weight. We also investigated the effects of Ibu administration on the serum lipid levels in db/db mice. Compared with the vehicle-treated db/db mice, the levels of CHO, TG, FFA, LDL-C, HDL-C, and ADPN in the serum of db/db mice treated with Ibu did not change significantly. Moreover, we noticed the serum glucose and insulin levels and glucose tolerance were not improved in Ibu-treated mice compared with vehicle-treated mice. This suggests that Ibu alone has no potential anti-obesity effect. These results are included in Fig. S9.

In addition, the solubility of BBR and BJ in water and octanol was also tested at 25°C. The solubility results are shown in Table 1 below. At 25°C, the solubility of BBR and BJ in water is not much different, but in octanol, the solubility of BJ is much higher than the solubility of BBR. Log P is the logarithmic value of the oil-water partition coefficient P, which refers to the partition equilibrium of undissociated molecules in the oil and water phases. The log P value is closely related to the water solubility, membrane permeability, *in vivo* ADME process, and affinity of the compound to the receptor and plays a vital role in the permeation of compounds through biological membranes. The log P 0-5 range is optimal for oral drug penetration by passive diffusion, with poor water solubility for high log P compounds and poor lipid permeability for low log P compounds¹⁻³. The log P value of BJ is well within the range of 0-5 compared to BBR, indicating that BJ has more lipid permeability than BBR. It also suggests that BJ can penetrate the cell membrane better than BBR in cell-related experiments to exert drug activity in cells.

Sample	concentration (nmol·mL ⁻¹)		log P
	H ₂ O	Octanol	
BBR	15.79±0.81	14.35±0.41	-0.041

BJ	6.02±0.35	220.39±0.73	1.564
----	-----------	-------------	-------

Table 1. The solubilities of BBR and BJ (n=3, 25°C). $\log P = \log ([\text{organic phase of compound}] / [\text{aqueous phase of compound}])$

References:

1. Lipinski CA, Lombardo F, Dominy BW, Feeney PJ. Experimental and computational approaches to estimate solubility and permeability in drug discovery and development settings. *Adv Drug Deliv Rev.* 2001 Mar 1;46(1-3):3-26.
2. Leeson PD, Bento AP, Gaulton A, Hersey A, Manners EJ, Radoux CJ, Leach AR. Target-Based Evaluation of "Drug-Like" Properties and Ligand Efficiencies. *J Med Chem.* 2021 Jun 10;64(11):7210-7230.
3. Tinworth CP, Young RJ. Facts, Patterns, and Principles in Drug Discovery: Appraising the Rule of 5 with Measured Physicochemical Data. *J Med Chem.* 2020 Sep 24;63(18):10091-10108.

2. The study shows overexpression experiments, but not any gene knockdown. It could be informative to perform experiments involving TBK1 and IKK ϵ knockdown. If the effects of BJ/BBR are blunted by lack of TBK1/IKK ϵ , this would be stronger evidence to suggest that BJ/BBR act through these kinases.

Response: We thank the Reviewer for the recommendations as the new experiments greatly improve the quality of our work. We reduced the TBK1 or IKK ϵ level in 3T3-L1 adipocytes by siRNA-mediated knockdown. The results showed that TBK1 knockdown or BJ alone increased the expression of p-AMPK α and UCP1 in 3T3-L1 adipocytes and promoted mitochondrial biogenesis. Compared with BJ treatment alone, the expression levels of p-AMPK α and UCP1 were not significantly changed in TBK1-knockdown 3T3-L1 adipocytes treated with BJ. Similarly, IKK ϵ knockdown or BJ alone increased the expression of UCP1 in 3T3-L1 adipocytes. However, compared with BJ treatment alone, the UCP1 expression was not significantly changed in IKK ϵ -knockdown 3T3-L1 adipocytes treated with BJ. Further, BJ alone treatment significantly increased cAMP accumulation, glycerol release, and UCP1 expression in ISO-stimulated 3T3-L1 adipocytes. Compared with BJ treatment alone, BJ did not

significantly increase the above changes in IKK ϵ -knockdown 3T3-L1 adipocytes in response to ISO stimulation. Taken together, our data suggest that the effects of BJ were blunted by lack of TBK1 or IKK ϵ . This further strengthens the evidence that BJ regulates adipocytes through TBK1 and IKK ϵ . These results are included in the new Fig. 9.

3. Lines 50-51. Authors claim that clinical studies have shown upregulation of IKK ϵ and TBK1 in patients with NAFLD by citing work by Oral et al. (reference 17). However, the paper cited did not measure IKK ϵ and TBK1 in human patients, but rather tested an IKK ϵ /TBK1 inhibitor in obese patients. While the abstract states that TBK1 and IKK ϵ are increased with obesity, the introduction explains that this was previously shown in mice, not humans. The authors should correct the wording of their statement.
Response: We apologize for the mistake. We have corrected our statement in the manuscript (lines 52-53).

4. Lines 91-92. Reference 26 seems to be out of place as the cited paper does not appear to include human subjects. Perhaps it should be reference 27 instead?
Response: The Reviewer is quite correct, and we apologize for the mistake. The correct reference is Reference 27. We have deleted Reference 26 from the manuscript and revised the Reference for this sentence (lines 98 and 930-931).

5. As the methods section is at the end of the paper, abbreviations for the different buffer solutions (SGF, FaSSIF-V1, FeSSIF-V1) should be defined in the results section where they first appear (line 168).
Response: Many thanks for the suggestion. We have now defined abbreviations for different buffer solutions (SGF, Fassif-V1, FESSIf-V1) in the results section where they first appear (lines 176-177).

6. Figure 4d shows “Fat Mass (%)”. The figure legend states that this is the “weight of adipose tissue” but it is not clear what this actually is. Is it the sum of the adipose tissue

weights at termination divided by the body mass? If so, it must include tissues other than BAT, iWAT and eWAT because the numbers in Fig 4c do not appear to add up. Or is it a non-invasive measurement of total body fat mass such as DEXA or EchoMRI? If so, this should be described in the methods and should not be called “weight of adipose tissue” since it would also include lipid content from other tissues.

Response: Thank you for pointing out this issue. Figure 4d shows the percentage of fat depots weight (brown fat, inguinal subcutaneous fat, epididymal fat, perirenal fat, and mesenteric fat depot) in the body weight of mice. We have described this in methods (lines 730-733), and modified it in the figure legend (lines 284-286).

7. Figure 5f. The ITT data are quantified by measuring the area under the curve. But these changes are exaggerated due to differences in fasting glucose. The initial slope of the curve (0-15 min) would be a better indicator of insulin sensitivity.

Response: Thank you for pointing out this issue. We have recalculated the initial slope of the ITT curve and added figure S10 to include the quantified data.

8. “Vehicle” is spelt incorrectly in Figure 5g.

Response: We apologize for our careless mistake. We have respelled it correctly in Figure 5g.

9. In Figure 5j, authors show only four representative images of F4/80 staining and claim that the proportion of F4/80-positive macrophages around adipocytes was decreased in BJ- and BBR-treated mice. The authors should support their argument with quantification of several random fields of view from each section for each mouse. The methods section (4.4.12) also does not state how the F4/80-positive macrophages were counted or quantified. In fact, the authors have stated that they used biotinylated secondaries but no streptavidin-HRP or substrate, and that they performed DAPI staining on IHC samples then observed with a fluorescence microscope (lines 685-688). This section should be checked and corrected.

Response: Thank you for pointing out this issue. We have added figure 5k to include

the quantified data. To quantify the F4/80-positive macrophages, three random fields per sample were analyzed by Image J software. The IHC Profiler plugin of ImageJ was used to calculate the F4/80 positive area (lines 764-766). We apologize for the description confusion in the methods section (4.4.12), and we have corrected it (lines 761-764).

10. Figure 7a-c. BJ appears to have stronger effects than BBR in vitro when both are administered at 10 μ M. It could be informative to discuss possible reasons for this phenomenon since it cannot be due to a difference in oral bioavailability (see comment 1).

Response: Many thanks for the constructive comments. Although BJ is a eutectic formed by hydrogen bonding between BBR and Ibu, the potential anti-obesity effect of Ibu was not found in our in vivo study. This suggests that Ibu does not affect the anti-obesity effect of BJ. By measuring the solubility of BBR and BJ in water and octanol at 25°C, we found that the log P value of BJ is better than that of BBR in the range of 0-5, indicating that BJ has stronger lipid permeability than BBR. This also indicates that BJ can penetrate cell membranes more effectively than BBR in vitro (see also a response to Point 1), which may partly explain why BJ shows stronger effects in vitro when both BJ and BBR are administered at the same dose.

11. Figure 7c. The authors claim that UCP1 expression is increased in BJ-treated 3T3-L1 cells compared to both vehicle and BBR, but there are only a few representative IF images shown. Was any fluorescence intensity quantification performed on these data? Also, MitoTracker is spelt incorrectly.

Response: Thank you for pointing out this issue. We have added figure 7d to include the fluorescence intensity quantification data. At least four images per section were acquired for quantification, and the mean intensity were evaluated using ImageJ. We apologize for our careless mistake. We have respelled it correctly in Figure 7c.

12. Lines 505-508. It is not clear what “the latter” is referring to. I initially thought “the

latter” was referring to 3T3-L1 adipocytes, but after multiple reads, I realised the intended meaning is that p-TBK1 upregulation may be due to feedback from AMPK activity. The sentence should be rewritten for clarity.

Response: Thank you for pointing out this issue. We have rewritten this sentence in the revised manuscript (line 574).

13. Line 514. “Adrenaline” should be replaced with “isoproterenol” for accuracy.

Response: Thank you for pointing out this issue. We have replaced “adrenaline” with “isoproterenol” in the revised manuscript (line 582).

14. Line 515. The authors did not show changes in mitochondrial biogenesis in vivo.

Response: Thank you for pointing out this issue. We have modified the sentence according to this comment (line 583).

15. Authors state that the One-Way ANOVA was performed to assess changes between groups. However, they do not state whether normality or homogeneity of variance was tested. If the data fail the normality test by a meaningful amount, a non-parametric test should be used (or the data should be transformed e.g., log-transformed to achieve normality). If homogeneity of variance is not true, a Welch’s ANOVA should be performed instead.

Response: Thank you for pointing out this issue. We have re-analyzed the data and modified the statistical method in the “quantification and Statistical Analysis” section as follows:

Comparisons of means in two groups were made using an unpaired t-test. Ordinary one-way ANOVA was used for intergroup analysis, followed by Tukey post-hoc analysis. Normality and homogeneity of variances were tested. When data did not comply with the homogeneity of variances, a t-test with Welch’s correction was performed, and Brown Forsythe & Welch one-way ANOVA with Dunnett’s multiple comparisons test was performed for comparison of multiple groups (lines 865-869).

16. Overall, the quality of the writing should be improved.

Response: Thank you for the comment. We have revised the sentences as you suggested and carefully edited the manuscript for English usage and grammar.

Reviewer #2 (Remarks to the Author):

This paper “A co-crystal berberine-ibuprofen improves obesity by inhibiting the protein kinases TBK1 and IKK ϵ ” studied the pharmacology effect of co-crystal BJ on obesity and its related metabolic diseases. It is very impressive that the effects on lipid metabolism, glucose metabolism, and inflammation in db/db mice were superior to berberine. However, the manuscript would be significantly improved by the following:

1. This article wanted to describe the effect of the co-crystal BJ, but it only compares with the effect of berberine alone, so I wonder whether ibuprofen has some effects on regulating obesity. How to exclude the influence of ibuprofen? We hope that the authors can make a more convincing statement in introduction, or give direct experimental pieces of evidence in results.

We thank the Reviewer for this constructive comment. To explore the potential impact of Ibu on obesity, we treated db/db mice with Ibu (150 mg/kg) or vehicle for 5 weeks. Ibu treatment did not affect body weight and fat depots weight. We also investigated the effects of Ibu administration on the serum lipid levels in db/db mice. Compared with the vehicle-treated db/db mice, the levels of CHO, TG, FFA, LDL-C, HDL-C, and ADPN in the serum of db/db mice treated with Ibu did not change significantly. Moreover, we noticed the serum glucose and insulin levels, as well as the glucose tolerance, were not improved in Ibu-treated mice compared with vehicle-treated mice. This suggests that Ibu alone has no potential anti-obesity effect. These results are included in Fig. S9.

2. Since BJ could promote the oral absorption of BBR and significantly enhance the bioavailability of BBR, what's the accumulation of BJ in adipose tissue compared with BBR alone?

Response: We thank the Reviewer for this comment. In line with the Reviewer's suggestion, we analyzed the accumulation of BBR in adipose tissue of db/db mice. As shown in the Figure 1 below, BBR was not detected in adipose tissue of db/db mice in each group by crushing, extracting, and HPLC analysis. This suggests that BJ or BBR does not accumulate. The better pharmacological activity of BJ compared with BBR is based on the improved bioavailability.

Figure 1. (a) The positions of the peaks of berberine standards in the HPLC assay. Analysis of berberine content in adipose tissue of each group Vehicle (b), BJ (75mg/kg) (c), BJ (150mg/kg) (d), and BBR (150mg/kg) (e).

3. Why the wild type group was missing when evaluate the effect of BJ on body weight,

energy expenditure, lipid, and glucose metabolism.

Response: We thank the Reviewer for this comment. In fact, when we analyzed the effects of BJ on db/db mice, we also analyzed the changes of body weight, energy expenditure, lipid, and glucose metabolism of wild type mice. These changes were significantly different in the wild type group compared with db/db mice. Meanwhile, we consulted lots of previous studies about db/db mice¹⁻⁶, which did not have a wild type group, as shown in the References below. Considering the genetic differences between WT and db/db mice, these changes were not of comparative significance between the two groups, and we removed these changes in WT mice. We have now added these data to the Figure 2 below.

Figure 2. (a) Representative images of wild type (WT) and db/db mice and their eWAT. (b-c) Body weight of WT and db/db mice (n=10 per group). (d) Changes of average food intake in each group during 5 weeks of treatment (n=6 per group). (e-h) Oxygen consumption (VO_2) (e-f) and carbon dioxide release (VCO_2) (g-h) of WT and db/db mice within 24 hours after treatment with BJ or BBR for 5 weeks (n=6 per group). (i) Respiratory exchange ratio (VCO_2/VO_2) over a 24-h period in each group (n=6 per group). (j-k) Spontaneous horizontal activity of WT and db/db mice over a 24-h period (n=6 per group). (l) Representative HE staining images of epididymal adipose tissue

(scale bar = 100 μ m) and adipocyte size quantification. (m) Weight of BAT, iWAT and eWAT in different groups (n=10 per group). (n) Percentage of fat depots weight to the body weight (n=10 per group). (o-t) Serum levels of CHO, TG, FFA, LDL-C, HDL-C and ADPN in the fasted state (n=6 per group). (u) Weekly fasting glucose levels of WT and db/db mice (n=6 per group). (v) Fasting serum insulin levels in WT and db/db mice after 35 days of treatment (n=6 per group). (w-x) GTT and the area under the curve (AUC) (n=6 per group). (y-z) ITT and the AUC (n=6 per group). Data are expressed as mean \pm SD. NS = not significant; *P < 0.05, **P < 0.01, ***P < 0.001.

References:

1. Zhang Z, Zhang H, Li B, Meng X, Wang J, Zhang Y, Yao S, Ma Q, Jin L, Yang J, Wang W, Ning G. Berberine activates thermogenesis in white and brown adipose tissue. *Nat Commun.* 2014 Nov 25;5:5493.
2. Song KH, Lee SH, Kim BY, Park AY, Kim JY. Extracts of *Scutellaria baicalensis* reduced body weight and blood triglyceride in db/db Mice. *Phytother Res.* 2013 Feb;27(2):244-50.
3. Feng L, Luo H, Xu Z, Yang Z, Du G, Zhang Y, Yu L, Hu K, Zhu W, Tong Q, Chen K, Guo F, Huang C, Li Y. Bavachinin, as a novel natural pan-PPAR agonist, exhibits unique synergistic effects with synthetic PPAR- γ and PPAR- α agonists on carbohydrate and lipid metabolism in db/db and diet-induced obese mice. *Diabetologia.* 2016 Jun;59(6):1276-86.
4. Everard A, Matamoros S, Geurts L, Delzenne NM, Cani PD. *Saccharomyces boulardii* administration changes gut microbiota and reduces hepatic steatosis, low-grade inflammation, and fat mass in obese and type 2 diabetic db/db mice. *mBio.* 2014 Jun 10;5(3):e01011-14.
5. Maessen DE, Brouwers O, Gaens KH, Wouters K, Cleutjens JP, Janssen BJ, Miyata T, Stehouwer CD, Schalkwijk CG. Delayed Intervention With Pyridoxamine Improves Metabolic Function and Prevents Adipose Tissue Inflammation and Insulin Resistance in High-Fat Diet-Induced Obese Mice. *Diabetes.* 2016 Apr;65(4):956-66.
6. Xue M, Zhang L, Yang MX, Zhang W, Li XM, Ou ZM, Li ZP, Liu SH, Li XJ, Yang SY. Berberine-loaded solid lipid nanoparticles are concentrated in the liver and

ameliorate hepatosteatosis in db/db mice. *Int J Nanomedicine*. 2015 Aug 5;10:5049-57.

4. In line 86, “oral BBR (1000 mg/d or 15000 mg/d) could reduce body weight, inflammation...”, 15000 mg/d of BBR is far more than the usual oral dosage. Please check it.

Response: Thank you for pointing out this issue. We apologize for our mistake. The oral dosage of BBR is 1500 mg/d (500 mg three times per day). We have modified the sentence according to this comment (line 90).

5. It is confused that BJ could both decrease the expression of TBK1 at mRNA level and inhibit the phosphorylation of TBK1 protein, since they couldn't occur simultaneously.

Response: We apologize for our lack of clarity. AMPK is a direct substrate of TBK1, and activation of AMPK in turn indirectly increases TBK1 activity by mediating the downstream Unc-51 like autophagy activating kinase 1 (ULK1). BJ increased TBK1 phosphorylation at Ser172, possibly due to an indirect promotion of AMPK activation (lines 370-371).

6. In Figure 8e and 8f, the results of Immunofluorescence staining for P-TBK1 and AMPK were lost.

Response: We thank the Reviewer for this comment. The results are included in the new Supplementary Figure S11e-h.

7. The manuscript requires extensive editing for English grammar and sentence structure.

Response: Thank you for the comment. We have revised the sentences as you suggested and carefully edited the manuscript for English usage and grammar.

REVIEWERS' COMMENTS:

Reviewer #1 (Remarks to the Author):

The authors have performed extensive revisions and further experiments that address the reviewer comments very well. The manuscript is now quite strong and the conclusions are well-supported by evidence.

I have only a few minor comments for editorial corrections.

- The first sentence of the abstract (lines 20-21) is still grammatically incorrect. "Anti-inflammatory" is not a noun (nor a disease). It's the very first sentence that readers will see so I think it's best if this is corrected.
- Lines 176-177. Thank you for defining the abbreviations here. However, the closing brackets should be repositioned to after "V1".
- Legend of Fig 7 and Fig 8: change "Mitotracter" to "MitoTracker".
- Thank you for the extensive knockdown experiments performed for Fig 9. When describing these results in the text, the authors frequently compare to "BJ alone". Perhaps they might consider changing the wording to also compare siRNA knockdown to siRNA knockdown + BJ. The lack of additional effect of BJ in the context of TBK1 or IKKe knockdown provides evidence that BJ is likely exerting its effects through these kinases.
- There are other minor grammatical corrections that could be made, but regardless, the article is easy to understand as it is.

Reviewer #2 (Remarks to the Author):

The authors answered all questions.

Reviewers' comments:

Reviewer #1 (Remarks to the Author):

The authors have performed extensive revisions and further experiments that address the reviewer comments very well. The manuscript is now quite strong and the conclusions are well-supported by evidence.

I have only a few minor comments for editorial corrections.

1. The first sentence of the abstract (lines 20-21) is still grammatically incorrect. “Anti-inflammatory” is not a noun (nor a disease). It’s the very first sentence that readers will see so I think it’s best if this is corrected.

2. Lines 176-177. Thank you for defining the abbreviations here. However, the closing brackets should be repositioned to after "V1".

3. Legend of Fig 7 and Fig 8: change “Mitotracter” to “MitoTracker”.

4. Thank you for the extensive knockdown experiments performed for Fig 9. When describing these results in the text, the authors frequently compare to “BJ alone”. Perhaps they might consider changing the wording to also compare siRNA knockdown to siRNA knockdown + BJ. The lack of additional effect of BJ in the context of TBK1 or IKKe knockdown provides evidence that BJ is likely exerting its effects through these kinases.

5. There are other minor grammatical corrections that could be made, but regardless, the article is easy to understand as it is.

Reviewer #2 (Remarks to the Author):

The authors answered all questions.

Point-by-Point response

Reviewer #1 (Remarks to the Author):

The authors have performed extensive revisions and further experiments that address the reviewer comments very well. The manuscript is now quite strong and the

conclusions are well-supported by evidence.

I have only a few minor comments for editorial corrections.

1. The first sentence of the abstract (lines 20-21) is still grammatically incorrect. “Anti-inflammatory” is not a noun (nor a disease). It’s the very first sentence that readers will see so I think it’s best if this is corrected.

Response: We apologize for the mistake. We have changed “anti-inflammatory” to “inflammation-associated diseases” in the first sentence of the abstract (line 21).

2. Lines 176-177. Thank you for defining the abbreviations here. However, the closing brackets should be repositioned to after "V1".

Response: Many thanks for the suggestion. According to your suggestion, we have redefined these abbreviations by repositioning the closing bracket after "V1" (lines 168-169).

3. Legend of Fig 7 and Fig 8: change “Mitotracter” to “MitoTracker”.

Response: We apologize for the mistake. We have changed "Mitotracter" to "MitoTracker" in the legends of Fig 7 and Fig 8 (lines 974, 975, 989 and 991).

4. Thank you for the extensive knockdown experiments performed for Fig 9. When describing these results in the text, the authors frequently compare to “BJ alone”. Perhaps they might consider changing the wording to also compare siRNA knockdown to siRNA knockdown + BJ. The lack of additional effect of BJ in the context of TBK1 or IKKe knockdown provides evidence that BJ is likely exerting its effects through these kinases.

Response: Many thanks for the suggestion. According to your suggestion, the comparison between BJ alone and siRNA knockout + BJ was replaced by that between siRNA knockout and siRNA knockout + BJ when we described the results in Fig 9 (lines 398-400, 404-406 and 408-410).

5. There are other minor grammatical corrections that could be made, but regardless,

the article is easy to understand as it is.

Response: We thank you for noting these grammatical mistakes. The manuscript has been sent for language editing to correct grammatical errors thoroughly.